# Two Late Cretaceous sauropods reveal titanosaurian dispersal across South America

E. Martín Hechenleitner 1,2✉, Léa Leuzinger1,3, Agustín G. Martinelli 4, Sebastián Rocher 5, Lucas E. Fiorelli 1, Jeremías R. A. Taborda6 & Leonardo Salgado7

South American titanosaurians have been central to the study of the evolution of Cretaceous sauropod dinosaurs. Despite their remarkable diversity, the fragmentary condition of several taxa and the scarcity of records outside Patagonia and southwestern Brazil have hindered the study of continental-scale paleobiogeographic relationships. We describe two new Late Cretaceous titanosaurians from Quebrada de Santo Domingo (La Rioja, Argentina), which help to fill a gap between these main areas of the continent. Our phylogenetic analysis recovers both new species, and several Brazilian taxa, within Rinconsauria. The data suggest that, towards the end of the Cretaceous, this clade spread throughout southern South America. At the same locality, we discovered numerous accumulations of titanosaurian eggs, likely related to the new taxa. With eggs distributed in three levels along three kilometres, the new site is one of the largest ever found and provides further evidence of nesting site philopatry among Titanosauria.

[1] Centro Regional de Investigaciones Científicas y Transferencia Tecnológica de La Rioja (CRILAR), Provincia de La Rioja, UNLaR, SEGEMAR, UNCa, CONICET, Entre Ríos y Mendoza s/n (5301), Anillaco, La Rioja, Argentina. [2] Instituto de Biología de la Conservación y Paleobiología (IBICOPA), DACEFyN-UNLaR, 5300 La Rioja, Argentina. [3] Laboratorio de Paleontología de Vertebrados, Departamento de Ciencias Geológicas, Facultad de Ciencias Exactas y Naturales, Pabellón II, Universidad de Buenos Aires, Intendente Güiraldes 2160, Ciudad Universitaria (C1428EGA), Buenos Aires, Argentina. [4] CONICET-Sección Paleontología de Vertebrados, Museo Argentino de Ciencias Naturales "Bernardino Rivadavia", Av. Ángel Gallardo 470, C1405 DJR Buenos Aires, Argentina. [5] Instituto de Geología y Recursos Naturales, Centro de Investigación e Innovación Tecnológica, Universidad Nacional de La Rioja (INGeReN-CENIIT-UNLaR), Av. Gob. Vernet y Apóstol Felipe, 5300 La Rioja, Argentina. [6] Centro de Investigaciones en Ciencias de la Tierra (CICTERRA), Universidad Nacional de Córdoba, CONICET, FCEFyN, Vélez Sarsfield 1611, Ciudad Universitaria, X5016GCA Córdoba, Argentina. [7] Instituto de Investigación en Paleobiología y Geología, Universidad Nacional de Río Negro-CONICET, Av. Presidente Julio A. Roca 1242, 8332 General Roca, Río Negro, Argentina. ✉email: emhechenleitner@gmail.com

itanosaurian sauropods are a group of large, long-necked, herbivorous dinosaurs with a complex evolutionary history[1–6]. During the Late Cretaceous, they underwent an extensive evolutionary radiation worldwide. Most of their record in South America is restricted to Argentine Patagonia (e.g., Neuquén, Golfo San Jorge and Austral basins) and the Bauru Basin of SW Brazil[7–9] (Fig. 1a). Some studies have attempted to establish paleobiogeographic links between these regions[10,11], although there are remarkable faunistic differences between Patagonian and Brazilian titanosaurians[12–15]. Similarly, other contemporaneous tetrapods, such as pleurodiran turtles and notosuchian mesoeucrocodylians, also show heterogeneous distributions[16,17].

By the Late Cretaceous, vast regions of South America remained flooded by epicontinental seas[18], and although there are high-rank taxonomic similarities, the evidence of eventual connections between northern and southern terrestrial faunas are still scarce. The ubiquity of the clade Titanosauria in a geographically intermediate area is validated by the occurrence of the saltasaurids *Yamanasaurus* from Ecuador[19] and *Saltasaurus*[20]—plus a putative record of *Neuquensaurus*[21]—from NW Argentina (Fig. 1a), along with fragmentary accounts of sauropod dinosaurs in the latter region. However, saltasaurids have not been documented so far in the Bauru Basin nor other units in Brazil[11,22], and the non-saltasaurid specimens in NW Argentina are too fragmentary[23] to allow determination of paleobiogeographic relationships. In addition to saltasaurids, the other high-level clade amongst titanosaurians is the Colossosauria, recently stem-based defined as the most inclusive clade containing *Mendozasaurus* but not *Saltasaurus*, nor *Epachthosaurus*[9]. It includes the subclades Rinconsauria and Lognkosauria (plus a few related taxa), whose taxonomic composition has fluctuated over the years[2–4]. The fossil record of colossosaurians has, so far, a disparate distribution, with most of its members reported in Patagonia and SW Brazil.

Herein, we report the discovery of new dinosaurs from the Upper Cretaceous red beds of the Quebrada de Santo Domingo locality (QSD) in the Andes of La Rioja, NW Argentina (Fig. 1b). We recovered three partial skeletons that belong to two new derived titanosaurian dinosaur species (Fig. 1c, d) in different stratigraphic positions of the Ciénaga del Río Huaco Formation. Moreover, we found titanosaurian egg clutches and eggshells in an intermediate stratigraphic position, distributed in three levels. With an overwhelming abundance of eggs, QSD is one of the largest nesting sites documented worldwide. The results of our phylogenetic analysis incorporating the two new taxa suggest that they have Patagonian and Brazilian affinities, reinforcing the hypothesis of a close relationship between the titanosaurian sauropod faunas from northern and southern South America during the Late Cretaceous.

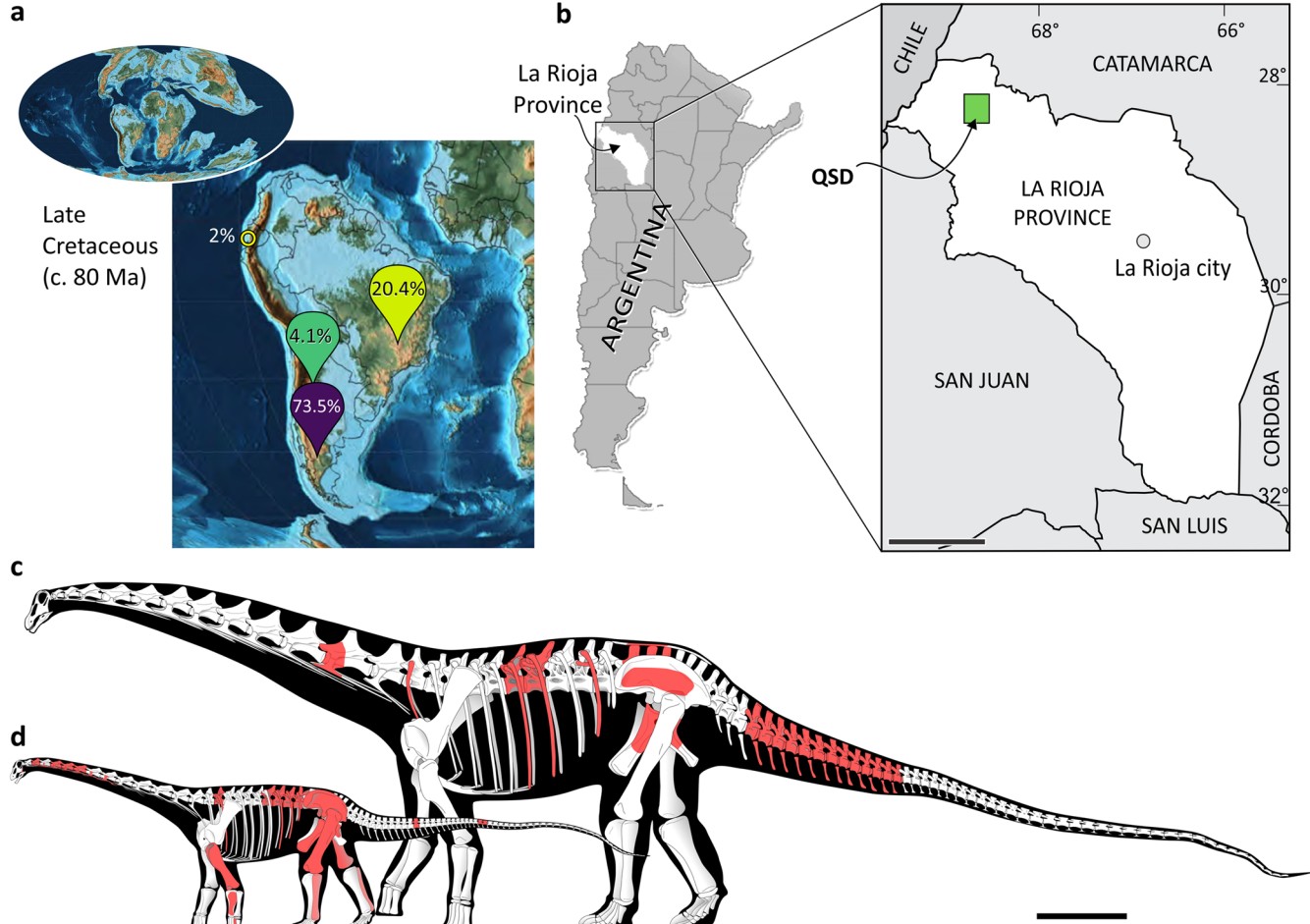

**Fig. 1 Titanosaurian record in South America, map of the study area and skeletal reconstructions of the new titanosaurian species. a** Percentage diversity of Cretaceous titanosaurian sauropods in three main regions of South America: Patagonia (purple), NW Argentina (green), and SW Brazil (yellow) (Supplementary Table 3). The yellow ring corresponds to the record of the saltasaurid titanosaurian *Yamanasaurus* in Ecuador. Map modified from Scotese[17]. **b** Location of the discoveries. **c** *Punatitan coughlini* gen. et sp. nov. **d** *Bravasaurus arrierosorum* gen. et sp. nov. Preserved elements are coloured in red in **c**, **d**. Scale bar: 100 km in **b**, and 1 m in **c**, **d**.

## Results

### Systematic palaeontology.

Sauropoda Marsh, 1878
Titanosauria Bonaparte and Coria, 1993
Colossosauria González Riga et al., 2019
*Punatitan coughlini* gen. et sp. nov.

**Etymology.** 'Puna' is the local name that distinguishes the oxygen-depleted atmosphere typical of the high Andes, and 'coughlini' refers to the geologist Tim Coughlin, who reported the first dinosaur fossils in the area.

**Holotype.** CRILAR-Pv 614 (Paleovertebrate Collection of Centro Regional de Investigaciones Científicas y Transferencia Tecnológica de La Rioja, Argentina), partial skeleton composed of the anterior portion of posterior cervical vertebra (likely C12), two middle dorsal vertebrae (likely D6–D7), partial sacrum, 13 articulated caudal vertebrae (some with articulated haemal arches), right pubis, left ischium, and several dorsal ribs.

**Horizon and type locality.** Sandstone levels 170 m above the base of the Ciénaga del Río Huaco Formation (Campanian-Maastrichtian) at QSD, La Rioja, NW Argentina (Geological Setting in Supplementary Information).

**Diagnosis.** A medium-sized titanosaurian sauropod characterised by the following combination of features (autapomorphies marked with an asterisk): (1) middle dorsal vertebrae (likely D6–D7) with anterior and posterior spinodiapophyseal laminae (spdl) forming wide and flat surface, between aliform and transverse processes*; (2) accessory posterior centrodiapophyseal lamina (apcdl) crossed over by the posterior centroparapophyseal (pcpl) lamina, forming a X-shaped intersection in D6–D7; (3) pcpl reaches the bottom of posterior centrodiapophyseal lamina (pcdl) in D6–D7*; (4) extra-depression ventrally to intersection of pcpl and apcdl in D6–D7*; (5) deep postzygodiapophyseal centrodiapophyseal fossa (pocdf) in D6–D7; (6) neural spine of D6 tapering dorsally, forming an inverted-"V" profile in anterior/posterior view; (7) caudal transverse processes persist beyond Ca15; (8) slightly anteriorly inclined neural spines in anterior-middle caudal vertebrae (Ca5–6 to Ca10); and (9) distally expanded prezygapophyses in anterior-middle caudal vertebrae.

**Description and comparisons of *Punatitan*.** Most diagnostic features are in the axial skeleton of *Punatitan* (Fig. 2), allowing us to distinguish the new taxon from other titanosaurians. The holotype CRILAR-Pv 614 represents a medium-sized individual, larger than the holotypes of *Overosaurus*[24], *Saltasaurus*[25], *Neuquensaurus*[26,27], and *Trigonosaurus*[28], about the same size as the holotype of *Uberabatitan*[29], and smaller than *Aeolosaurus*[30], '*Aeolosaurus*'[11], *Mendozasaurus*[3] and giant taxa (e.g., *Argentinosaurus*, *Patagotitan*).

A cranial portion of a posterior cervical vertebra is only available (Fig. 2a, b). It may correspond to C12, based on *Overosaurus* and *Trigonosaurus* (MCT 1499-R[28]). The centrum is shorter dorsoventrally than it is wide transversely, with its anterior surface strongly convex. The base of the right parapophysis is level with the ventral border of the centrum and ventrally delimits the deeply concave lateral surface of the centrum. The prezygapophyses are anterolaterally projected and well separated from each other. Their anterior edge is placed slightly anterior to the level of the articular surface. Both are medially connected by a sharp interprezygapophyseal lamina (tprl) that forms an opened U-shaped edge in dorsal view. The right base of a rounded dorsomedially projected spinoprezygapophyseal lamina (sprl) is preserved. Although the

neural arch is incomplete, the position and development of the prezygapophyses, together with the position, orientation, and robustness of the sprl, suggest a wide and concave spinoprezygapophyseal fossa (sprf). Overall, the cervical vertebra of *Punatitan* is similar to that of most titanosaurians. The robust sprl is more similar to that of *Malawisaurus*[31], *Mendozasaurus*[3], *Futalognkosaurus*[32], and *Dreadnoughtus*[33] than to *Overosaurus*[24], in which the lamina is weakly developed, and the floor of the sprf is reduced. In *Trigonosaurus*[28] the sprl is also conspicuous but relatively short, thus defining a small sprf.

Two dorsal vertebrae are known for *Punatitan*, interpreted as D6 (Fig. 2c, d) and D7 (Fig. 2e), based on comparisons with *Overosaurus*[24] and *Trigonosaurus*[28] (e.g., the relative position of parapophysis and diapophysis, orientation of neural spine). The centra are opisthocoelous, almost as high as wide. Laterally, they show deep and partitioned pleurocoels that have tapering, acute caudal margins. They are located dorsally, near the neurocentral junction. The neural arches are fused to the centra, without a sign of suture.

The diapophyses are robust and well projected laterally, while the parapophyses are more anteriorly and slightly ventrally positioned, as occurs in middle dorsal vertebrae (e.g., D5–D7 of *Overosaurus*[24]). Below these processes, the neural arches are notably intricate, showing a broad, deeply excavated fossa (Fig. 2c) with a conspicuous asymmetry in both lateral sides, as seen in other sauropods (e.g., *Trigonosaurus*[28], *Lirainosaurus*[34]).

The pcdl and its anterior projection, the apcdl, plus the well-developed pcpl are the most conspicuous traits in the lateral aspects of these vertebrae (Fig. 2c), as seen in several titanosaurians, such as *Malawisaurus*[31], *Elaltitan*[35], *Overosaurus*[24], *Trigonosaurus*[28], and *Dreadnoughtus*[33]. The pcdl projects posteriorly to reach the posterodorsal border of the centrum. The apcdl projects anteriorly from the dorsal edge of this lamina, contacting the anterodorsal border of the centrum. The accessory lamina is crossed over by the pcpl, forming an X-shaped intersection that is evident on the right side of D6 and D7 (on left sides of both, the pcpl finishes when contacting the apcdl, forming a Y-shaped pattern). The pattern observed in D6–D7 of *Punatitan* is roughly observed in D7 of *Overosaurus*[24] (other dorsal vertebrae have no clear X-pattern) and *Petrobrasaurus*[36], but not in other titanosaurians such as *Malawisaurus*[31], *Elaltitan*[35], *Trigonosaurus*[28], *Lirainosaurus*[34], and *Dreadnoughtus*[33]. Conspicuously, these laminae define deep fossae in *Punatitan*. The deep, subtriangular fossa, dorsally delimited by the pcdl and apcdl is identified as posterior centrodiapophyseal fossa (pcdl-f)[33]. It is deeper in *Punatitan* than in *Overosaurus*[24], *Trigonosaurus*[28], *Muyelensaurus*[37], and *Dreadnoughtus*[33].

The anterior centroparapophyseal lamina (acpl) and pcpl project ventrally and posteroventrally, respectively, from the parapophysis. The pcpl is truncated on the left side of D6–D7 when touching the apcdl; consequently, on this side, the pcdl-f is much larger than on the right side. In both dorsal vertebrae, the acpl and pcpl also define a deep but small fossa.

The oval-shaped prezygapophyses are connected medially by transversely short tprl (Fig. 2e). They are detached from the diapophyseal body by a marked step that dorsally elevates their articular surface. In anterior view, the centroprezygapophyseal lamina (cprl) has a sharp border, and it widens dorsally. This lamina and the acpl define a deep fossa that faces anterolaterally. The sprl in these dorsal vertebrae are present as blunt structures that are poorly preserved. They connect the prespinal lamina (prsl) medially, without obstructing its path. A similar condition was inferred for *Barrosasaurus*[38], and a posterior dorsal vertebra referred as to *Trigonosaurus*[39], but they can correspond to accessory laminae rather than to the true sprl, which is usually seen in more anterior vertebrae[40].

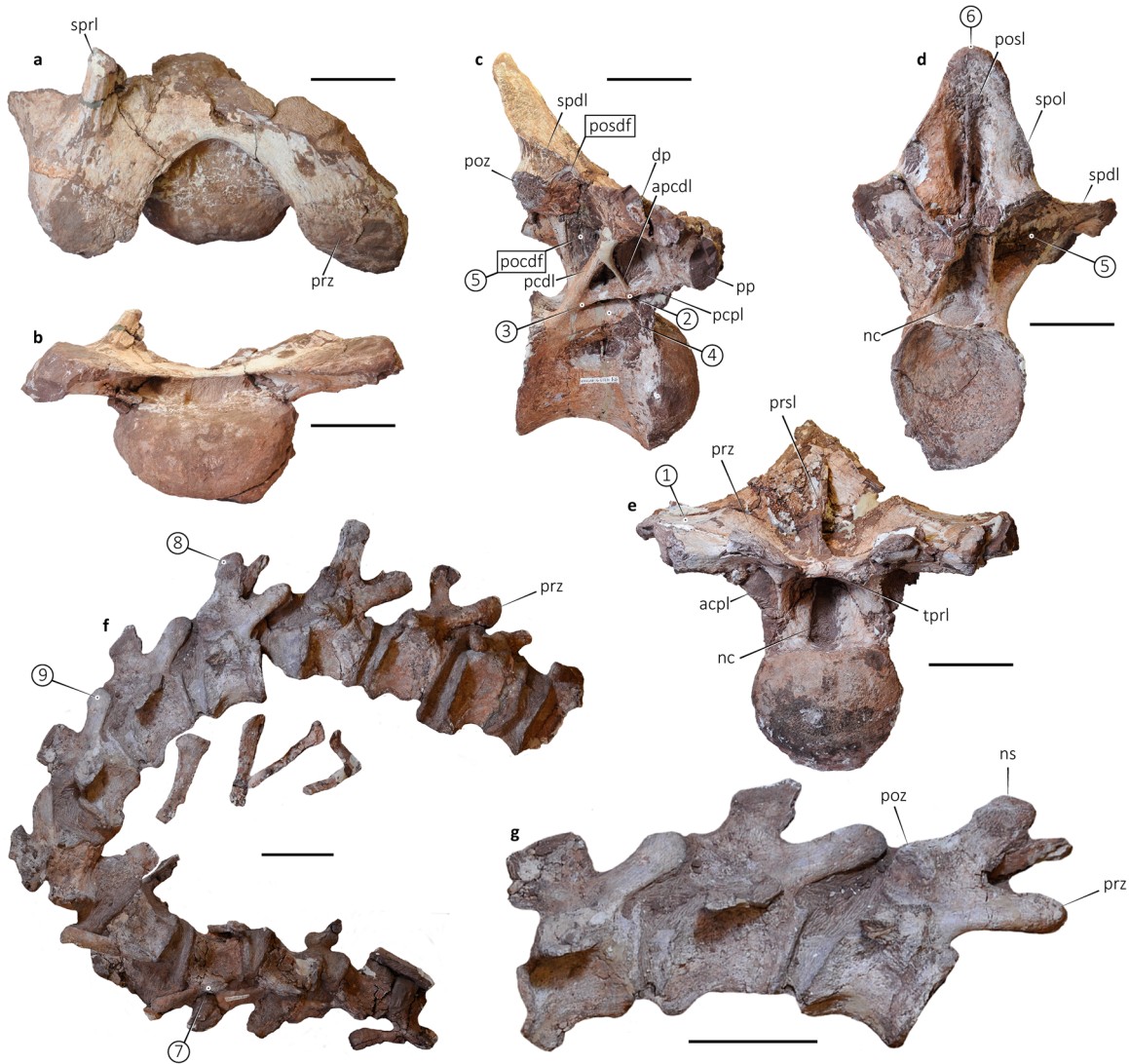

**Fig. 2 *Punatitan coughlini* gen. et sp. nov. (CRILAR-Pv 614). a**, **b** Cervical vertebra (C12) in dorsal **a** and anterior **b** views. **c**, **d** Dorsal vertebra (D6) in right lateral **c** and posterior **d** views. **e** Dorsal vertebra (D7) in anterior view. **f** Articulated series of caudal vertebrae (Ca5–Ca17). **g** Detail of Ca8–Ca12. acpl anterior centroparapophyseal lamina, apcdl accessory posterior centrodiapophyseal lamina, dp diapophysis, nc neural canal, ns neural spine, pcdl posterior centrodiapophyseal lamina. pcpl posterior centroparapophyseal lamina, pocdf postzygapophyseal centrodiapophyseal fossa, posdf postzygapophyseal spinodiapophyseal fossa, posl postspinal lamina, poz postzygapophysis, pp parapophysis, prsl prespinal lamina, prz prezygapophysis, spdl spinodiapophyseal lamina, spol spinopostzygapophyseal lamina, sprl spinoprezygapophyseal lamina, tprl interprezygapophyseal lamina. Circled numbers correspond to apomorphies numbered in the text. Measurements in Supplementary Table 1. Scale bars: 100 mm.

The postzygapophyses are higher than the lateral tip of the diapophysis in D6–D7, and there is no direct contact between the postzygapophyses and the diapophyses. Instead, there is a lamina that starts at the postzygapophysis and projects anterodorsally to connect to the spdl, closer to the base of the spine than to the base of the diapophysis. The homology of this lamina is debated[40,41]; it is here interpreted as the podl. This lamina is similar to the podl observed in dorsal vertebrae of *Malawisaurus*[31], *Choconsaurus* (D6?[42]) and *Dreadnoughtus* (D6?[33]), and its unusual connection with the spdl may be related to changes of the neural spine inclination and the relative position of the postzygapophyses and diapophyses in middle dorsal vertebrae[41]. At this point, this short podl delimits ventrally a very small postzygapophyseal spinodiapophyseal fossa (posdf), which faces laterally (Fig. 2c). A similar small fossa is present in the anteriormost dorsal of *Rapetosaurus*[43] and the mid-posterior dorsal of *Bonitasaura*[44]. It differs from the condition seen in *Lirainosaurus* and *Neuquensaurus*, in which the posdf is well developed and faces more posteriorly. The

postzygapophyses in D6 slope dorsally to the neural spine without a spinopostzygapophyseal lamina (spol), differing from the condition of *Dreadnoughtus*[33], *Mendozasaurus*[3] and *Elaltitan*[35], which have a sharp lamina. The centropostzygapophyseal lamina is also well developed, contacting the pcdl near the level of the neural canal. Both laminae define a large and deep pocdf.

The neural spine is complete in D6 of *Punatitan*. It is somewhat inclined posteriorly, with the tip extending as far posteriorly as the posterior border of the centrum (Fig. 2c). It is anteroposteriorly narrow and tapers dorsally. In anterior view, the contour of the tip is rounded, without any expansion, forming an inverted V-shaped profile, with a slightly sigmoid outline owing to the presence of aliform processes. The neural spine bears a prsl and a postspinal lamina (posl). The prsl is sharp in the basal half of the spine, separating two deep, wide fossae, laterally delimited by the prominent spdl. The posl is also sharp and expands over almost all the neural spine, delimiting two deep, narrow fossae, laterally bordered by the postzygapophyses, and

the aliform processes (Fig. 2d). The neural spine of D6 in *Punatitan* differs from that of most titanosaurians, which have expanded (e.g., *Dreadnoughtus*[33]) or squared (e.g., *Choconsaurus*[42], *Overosaurus*[24], *Trigonosaurus*[28]) neural spines.

The still unprepared sacrum of *Punatitan* is incomplete and will be described elsewhere. However, it was possible to observe an ossified supraspinous rod placed over the preserved neural spines (two or more). This structure is known for *Epachthosaurus*, *Malawisaurus*, and basal titanosauriforms[45].

The holotype of *Punatitan* also preserves 13 articulated caudal vertebrae as well as several haemal arches (Fig. 2f). The first preserved caudal possibly represents Ca5. As in most titanosaurians, these caudal vertebrae have strongly procoelous centra[1]. The centra are dorsoventrally tall, differing from the depressed centra of saltasaurines[25,46]. Their anterodorsal border is anteriorly displaced from the anteroventral one, resulting in an oblique profile in lateral view. They have slightly concave lateral surfaces, with transversely thin ventrolateral ridges that delimit a deeply concave ventral surface that is devoid of fossae. The internal tissue of the caudal centra is spongy, and the neural arches are apneumatic.

In the anterior caudal vertebrae, a suture is present above the base of the transverse processes (Fig. 2g). It forms a conspicuous ridge, which is not evident in related taxa, although it resembles the dorsal tuberosity described for *Baurutitan*[47], and also CRILAR-Pv 518c from Los Llanos, east La Rioja[23]. The neural arch of each caudal vertebra is situated over the anterior two-thirds of the centrum, and each is relatively tall with well-developed prezygapophyses and neural spines. The transverse processes are sub-triangular to laminar and gradually change from laterally to posterolaterally projected along the vertebral column. The prezygapophyses are long and project anterodorsally. The postzygapophyses contact the neural spine via a short spol and are located almost at the midline of the centra. This condition differs from the much more anteriorly placed postzygapophyses of the Patagonian *Aeolosaurus*[30]. The neural spine is rectangular in cross-section and anteroposteriorly longer than transversely wide (including prsl and posl). The spines are tall in the anterior caudal vertebrae and become shorter and square in the posterior ones. They also project slightly anteriorly, especially in Ca8–Ca10 (Fig. 2g). Some degree of anterior inclination of the neural spines is also reported for *Trigonosaurus*[28] and *Aeolosaurus*[30], contrasting with the most common condition amongst titanosaurians, i.e., vertical or posteriorly oriented neural spines (e.g., *Baurutitan*[47], *Dreadnoughtus*[33], *Saltasaurus*[25]). The available haemal arches are opened Y-shaped, with no expanded pedicels, as are those reported for other derived titanosaurians[48].

*Bravasaurus arrierosorum* gen. et sp. nov.

**Etymology.** *Bravasaurus*, referred to the Laguna Brava, a lake that gives name to the Laguna Brava Provincial Park, and *arrierosorum*, refers to the people who crossed the Andes carrying cattle during the 19th century.

**Holotype.** CRILAR-Pv 612, right quadrate and quadratojugal, four cervical, five dorsal, and three caudal vertebrae, few dorsal ribs, three haemal arches, left humerus, fragmentary ulna, metacarpal IV, partial left ilium with sacral ribs, right pubis, partial ischium, left femur, and both fibulae.

**Paratype.** CRILAR-Pv 613, isolated tooth, right ilium, right femur, and dorsal ribs.

**Horizon and type locality.** Sandstone levels 34 m above the base of the Ciénaga del Río Huaco Formation (Campanian-Maastrichtian) at QSD, La Rioja, NW Argentina (Geological Setting in Supplementary Information).

**Diagnosis.** A small-sized titanosaurian sauropod characterised by the following association of features (autapomorphies marked with an asterisk): (1) quadrate with articular surface entirely divided by medial sulcus*; (2) sprl forms conspicuous step between neural spine and prezygapophyses, in middle cervical vertebrae*; (3) strongly depressed centra (up to twice as wide as tall) in posterior dorsal vertebrae; (4) robust dorsal edge of pneumatic foramen in dorsal centra, forming prominent shelf that extends laterally, beyond the level of the ventral margin of the centum*; (5) posterior dorsal vertebrae with a rough posl, ventrally interrupted by middle spinopostzygapophyseal laminae (m.spol) that contact the postzygapophyses; (6) posterior dorsal vertebrae with small ventral spinopostzygapophyseal fossa (v.spof) delimited dorsally by the m.spol and ventrally by the interpostzygapophyseal lamina (tpol); (7) humerus with narrow midshaft, with midshaft/proximal width ratio of 0.36; (8) deltopectoral crest of the humerus expanded distally; (9) slender fibula (Robustness Index [RI][49] = 0.15); (10) distal condyle of the fibula transversely expanded, more than twice the midshaft breadth.

**Description and comparisons of *Bravasaurus*.** The holotype of *Bravasaurus* (Figs. 3 and 4), as well as the referred specimen, indicates a small-sized titanosaurian, much smaller than *Punatitan* (Fig. 1c, d) and other medium-sized sauropods, such as *Trigonosaurus*, *Overosaurus*, and *Bonitasaura*. Considering that both specimens could be adults (see below), they would be similar to *Neuquensaurus* or *Magyarosaurus*[50]. Cranial elements include partial right quadrate and quadratojugal (Fig. 3a, b). The quadrate is anteroventrally directed and bears part of the quadrate fossa. The articular surface for the mandible is transversely elongated. It shows two condyles that separate from each other by a longitudinal sulcus (Fig. 3b). The medial condyle is round, whereas the lateral is anteroposteriorly elongated. *Diplodocus*[51] also has a sulcus but restricted to the posterior region of the articular surface. Among titanosaurians, the articular surface of the quadrate has a kidney shape in *Nemegtosaurus* and *Quaesitosaurus*[52], with the sulcus restricted to its anterior portion. In *Bonitasaura*[53] and *Rapetosaurus*[54], the articular surface is not divided. The anterior process of the quadratojugal projects ventrally, whereas the posterolateral process barely extends ventrally, similar to *Nemegtosaurus*[52], and much less developed than in *Tapuiasaurus*[55] and *Sarmientosaurus*[4]. Unlike in these latter taxa, the posterolateral process reaches the articular condyle of the quadrate, which can only be seen behind (and not below) the quadratojugal in lateral view (Fig. 3a).

The holotype of *Bravasaurus* preserves cervical, dorsal, and caudal vertebrae. The neural arches of all elements are completely fused to their respective centra, which may indicate that it had reached somatic maturity before death[56–58].

We recovered four anterior-middle cervical vertebrae less than half a meter away from the cranial material. Three of them are articulated and associated with ribs. They are opisthocoelous, with sub-cylindrical and relatively elongated centra (Fig. 3c). The neural arches have low neural spines, as observed in *Rinconsaurus*[59] and *Uberabatitan*[29]. The diapophyses have posterior extensions, and the prezygapophyses are placed beyond the articular condyle of the centrum, as seen in the latter taxa. In *Bravasaurus* the postzygodiapophyseal lamina (podl) splits into a diapophyseal and a zygapophyseal segment, which become parallel with each other. Previous studies identified this feature as exclusive of *Uberabatitan*[13,29]. In derived titanosaurians, the neural spines contact the prezygapophyses via the sprl, which is straight or slightly curved ventrally in lateral view. In the anterior cervical vertebrae of few titanosaurians (e.g. *Saltasaurus*[25] and

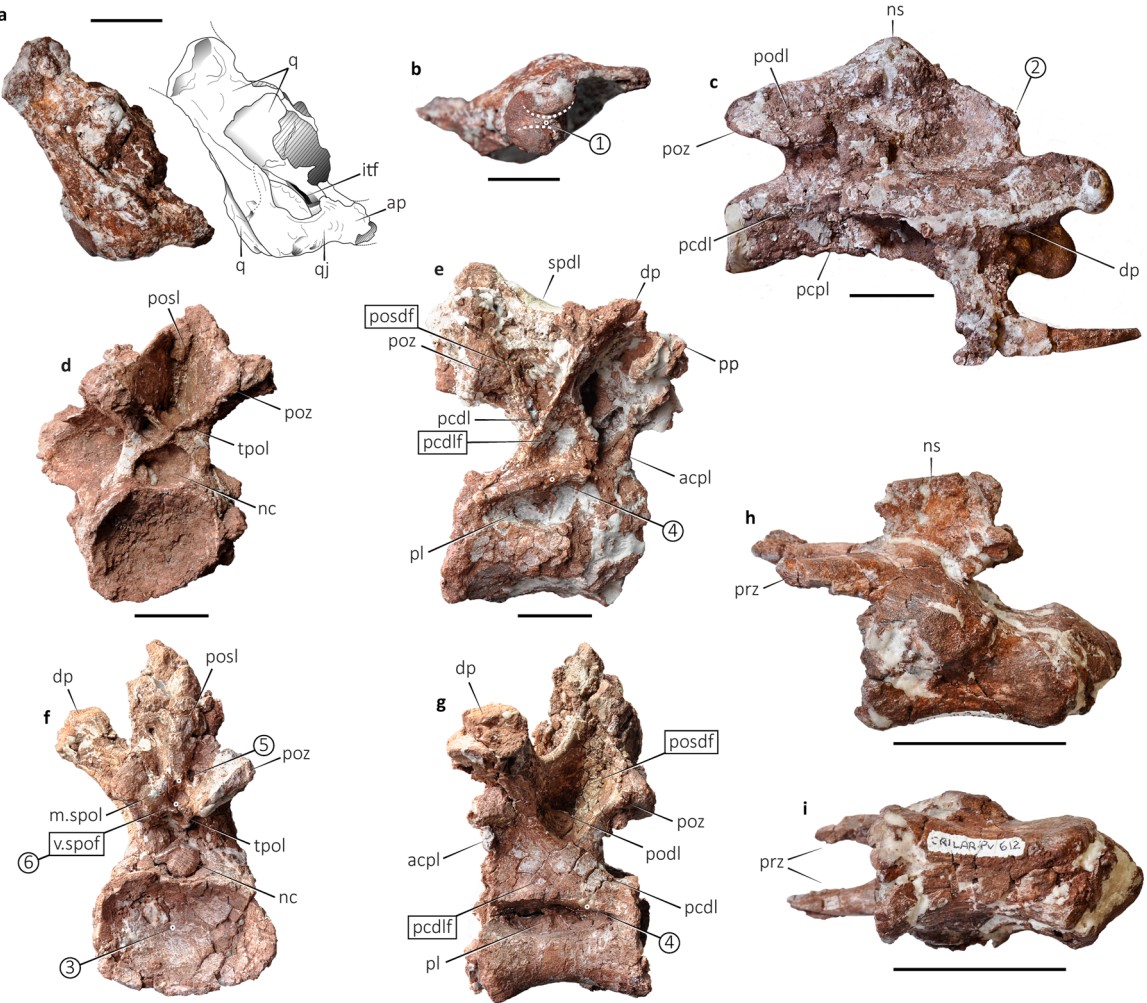

**Fig. 3 Axial elements of *Bravasaurus arrierosorum* gen. et sp. nov. (CRILAR-Pv 612). a, b** Quadrate and quadratojugal with interpretative drawing in right lateral **a**, and ventral **b** views (anterior to the right). **c** Middle cervical vertebra in right lateral view. **d** Anterior dorsal vertebra (D2) in posterior view. **e** Middle dorsal vertebra (D7) in right lateral view. **f–g** Posterior dorsal vertebra (D8) in posterior **f** and left lateral **g** views. **h, i** Middle caudal vertebra in left lateral **h** and ventral **i** views (anterior towards left). acpl anterior centroparapophyseal lamina, ap anterior projection, dp diapophysis, itf infratemporal fenestra, m.spol middle spinopostzygapophyseal lamina, nc neural canal, ns neural spine, pcdl posterior centrodiapophyseal lamina, pcdlf posterior centrodiapophyseal fossa, pcpl posterior centroparapophyseal lamina, pl pleurocoel, podl postzygodiapophyseal lamina, posdf postzygapophyseal spinodiapophyseal fossa, posl postspinal lamina, poz postzygapophysis, pp parapophysis, prz prezygapophysis, q quadrate, qj quadratojugal, spdl spinodiapophyseal lamina, tpol interpostzygapophyseal lamina, and v.spof ventral spinopostzygapophyseal fossa. Circled numbers correspond to apomorphies numbered in the text. Measurements in Supplementary Table 2. Scale bars: 10 mm in **a**, **b**, and 50 mm in **c–i**.

*Rocasaurus*[47]), the sprl curves dorsally, forming a step close to the prezygapophysis. This step disappears beyond the first cervical vertebrae but remains present in middle cervical vertebrae of *Bravasaurus* (C5?–C6?; Fig. 3c).

The dorsal vertebrae of *Bravasaurus* have relatively short, opisthocoelous centra (Fig. 3d–g). The well-developed pleurocoels are located just below the dorsal margin of the centrum, which forms a shelf that extends laterally, beyond the limits of the centrum, in middle and posterior dorsal vertebrae. Except for D10, the preserved dorsal centra are strongly dorsoventrally depressed (Fig. 3d, f), as in *Opisthocoelicaudia*[60], *Alamosaurus*[61], *Trigonosaurus*[28], and the "Series A" from Brazil[30]. The neural arches of the dorsal vertebrae are tall, but not as tall as in *Punatitan*, in which the pedicels are particularly long. The orientation of the preserved neural spines follows the same pattern as in other derived titanosaurians, i.e., vertical in anterior and posterior-most dorsal vertebrae, and inclined (as much as 40°) in middle dorsal vertebrae (e.g., *Trigonosaurus*[28]). The prsl

and posl are robust along their entire length (especially in the posterior dorsal vertebrae).

The anterior dorsal (D2) shows a low, laterally expanded neural arch (Fig. 3d). Although poorly preserved anteriorly, this vertebra exhibits a broad prespinal fossa with a weak prsl. It has rounded, ventrolaterally inclined postzygapophyses that reach the diapophyses though long podl. Medially, the postzygapophyses join each other by small laminae (tpol?) that intersect at the height of the dorsal edge of the neural canal. The junction between these laminae and the dorsal edge of the neural canal forms two small fossae, as seen in the posterior cervical vertebrae of *Overosaurus*[24]. The neural spine is relatively low, and the postspinal fossa is particularly deep compared with the other dorsal vertebrae. The posl is weak. On the lateral aspect, the pcdl and the apcdl are the most conspicuous laminae. The diapophysis is eroded, and the parapophysis is located on the centrum above the pleurocoel.

The middle dorsal (D7) shows a slightly higher neural arch than D2, and its neural spine is inclined posteriorly, beyond the

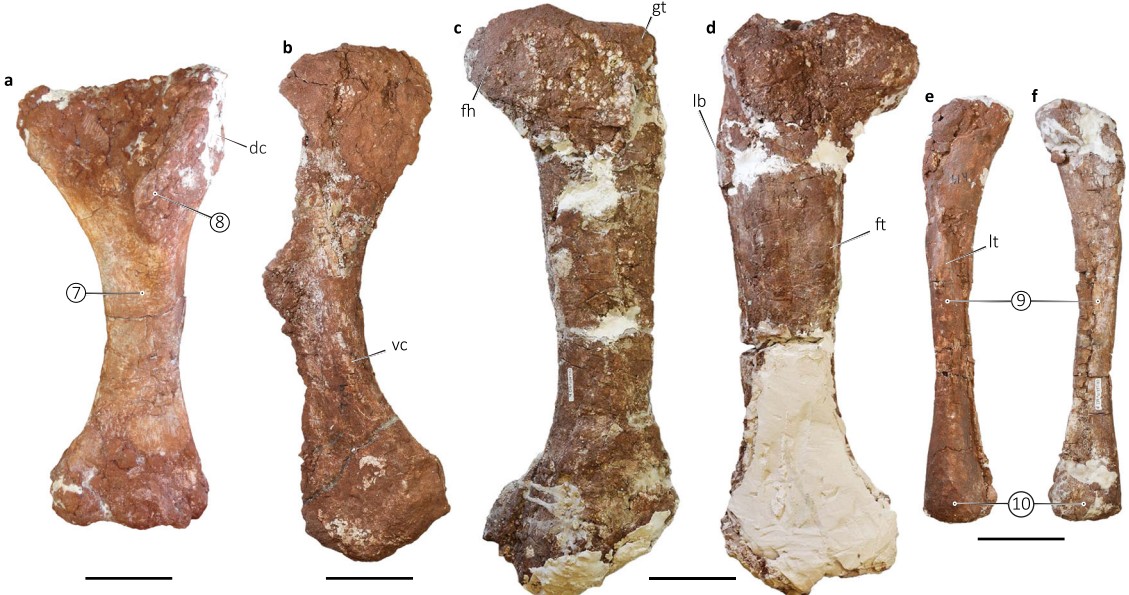

**Fig. 4 Appendicular elements of *Bravasaurus arrierosorum* gen. et sp. nov. (CRILAR-Pv 612). a** Left humerus in anterior view. **b** Right pubis in ventrolateral view. **c, d** Left femur in anterior **c** and posterior **d** views. **e, f** Right fibula in lateral **e** and medial **f** views. dc deltopectoral crest, fh femoral head, ft forth trochanter, gt greater trochanter, lb lateral bulge, lt lateral tuberosity, and vc ventral crest. Circled numbers correspond to apomorphies numbered in the text. Measurements in Supplementary Table 2. Scale bars: 100 mm.

posterior articular surface of the centrum (Fig. 3e). The parapophysis is missing, but the orientation of acpl and pcpl suggests a position slightly below and anterior to the diapophysis. In D8 and D10, a pair of m.spol interrupts the path of the posl, ventrally limiting a single, small fossa, here interpreted as v.spof (Fig. 3f). Its ventral limit corresponds to the tpol. A similar structure is present in *Lirainosaurus*[34]. The podl is present in all the posterior dorsal vertebrae (D8–D10).

The anterior and middle caudal vertebrae of *Bravasaurus* are procoelous. The centra are as tall dorsoventrally as they are wide transversely, without any concavities on their ventral surfaces (Fig. 3h, i). The anterior margin of the centra does not appear to be anteroventrally inclined, as occurs in *Punatitan, Overosaurus*[24], or *Aeolosaurus*[30]. The neural arches are on the anterior portion of the centra, as in most titanosaurians, and some other titanosauriforms (e.g., *Wintonotitan*[62]). The neural spines are laminar and vertically directed, while the prezygapophyses are short and anteriorly projected. Such morphology shows many similarities with *Rinconsaurus*[59] and *Muyelensaurus*[37], but even more so with the Brazilian *Trigonosaurus*[28] and *Uberabatitan*[13,29]. As for the centra, *Bravasaurus* differs from saltasaurines, in which they are depressed, with a ventral longitudinal hollow (e.g., *Saltasaurus*[25]). Nor do they possess the ventrolateral ridges (Fig. 3i) present in other titanosaurians such as *Aeolosaurus*[30], *Overosaurus*[24], and *Punatitan*. *Bravasaurus* also differs from the latter taxa by the orientation of the neural spine in the anterior caudal, which is vertical rather than anteriorly directed. None of the preserved caudal vertebrae shows signs of distal expansion in the prezygapophyses, as seen in *Punatitan*.

The morphology of the humerus is compatible with that of many colossosaurian titanosaurians. Its robustness is high (RI = 0.35), as in *Opisthocoelicaudia*[60], *Diamantinasaurus*[63], and *Savannasaurus*[64], much more than in *Rinconsaurus*[59] and *Muyelensaurus*[37]. The deltopectoral crest is markedly expanded distally (Fig. 4a), as in *Saltasaurus*[25], *Neuquensaurus*[27], *Opisthocoelicaudia*[60], and *Dreadnoughtus*[33]. All pelvic elements are represented in the holotype, although only the pubis (Fig. 4b)

allows comparisons. It is proximodistally elongate and less robust than in *Futalognkosaurus*[32] or *Opisthocoelicaudia*[60]. The distal end is markedly expanded, as in several derived forms (e.g., *Rapetosaurus*[43], *Bonitasaura*[44], *Muyelensaurus*[37]). The ilium of the specimen CRILAR-Pv 613 resembles the ilium of other derived titanosaurians, such as *Rapetosaurus* and *Bonatitan*[65]. The femur is straight, with the fourth trochanter placed at the proximal third (Fig. 4c, d), as in *Uberabatitan*[13], *Patagotitan*[2], *Bonitasaura*[44], and *Futalognkosaurus*[32], whereas in *Rinconsaurus*[59], *Muyelensaurus*[37], and *Diamantinasaurus*[63] it is located in the middle third. The humerus-to-femur length ratio in *Bravasaurus* is 0.75, similar to *Opisthocoelicaudia*, higher than *Neuquensaurus* and *Saltasaurus*, but lower than *Patagotitan* and *Epachthosaurus*. The fibula (Fig. 4e, f) markedly contrasts with the rest of the appendicular elements, as it is particularly gracile. Its distal condyle is transversely expanded, as observed in *Epachthosaurus*[66].

The known specimens of *Bravasaurus* indicate a small adult size. We estimate a body mass of 2.89 tons (2.17–3.61 tons, considering 25% error), based on a calibrated equation[67] (see "Methods" section). Estimates of <10 tons are few among titanosaurians. The European *Magyarosaurus* (750 kg), is interpreted as a case of insular dwarfism[50,68]. The mass of the European *Lirainosaurus* was less than two tons[50], whereas that of the Argentinean *Saltasaurus* and *Neuquensaurus* was five and six tons[2], respectively. Among colossosaurians, estimations for *Rinconsaurus* indicate just four tons[2] and at least some other genera (e.g., *Overosaurus*, *Trigonosaurus*, *Baurutitan*), lacking appendicular bones, are small-sized forms, slightly larger than *Bravasaurus*, based on their vertebral size.

**Phylogenetic analysis**. The result of our phylogenetic analysis nests *Punatitan* and *Bravasaurus* as derived titanosaurians in all most parsimonious trees. The topology of the strict consensus tree is similar to that obtained in previous studies using the same dataset[2,6], although some taxa, such as *Baurutitan* and *Trigonosaurus* show

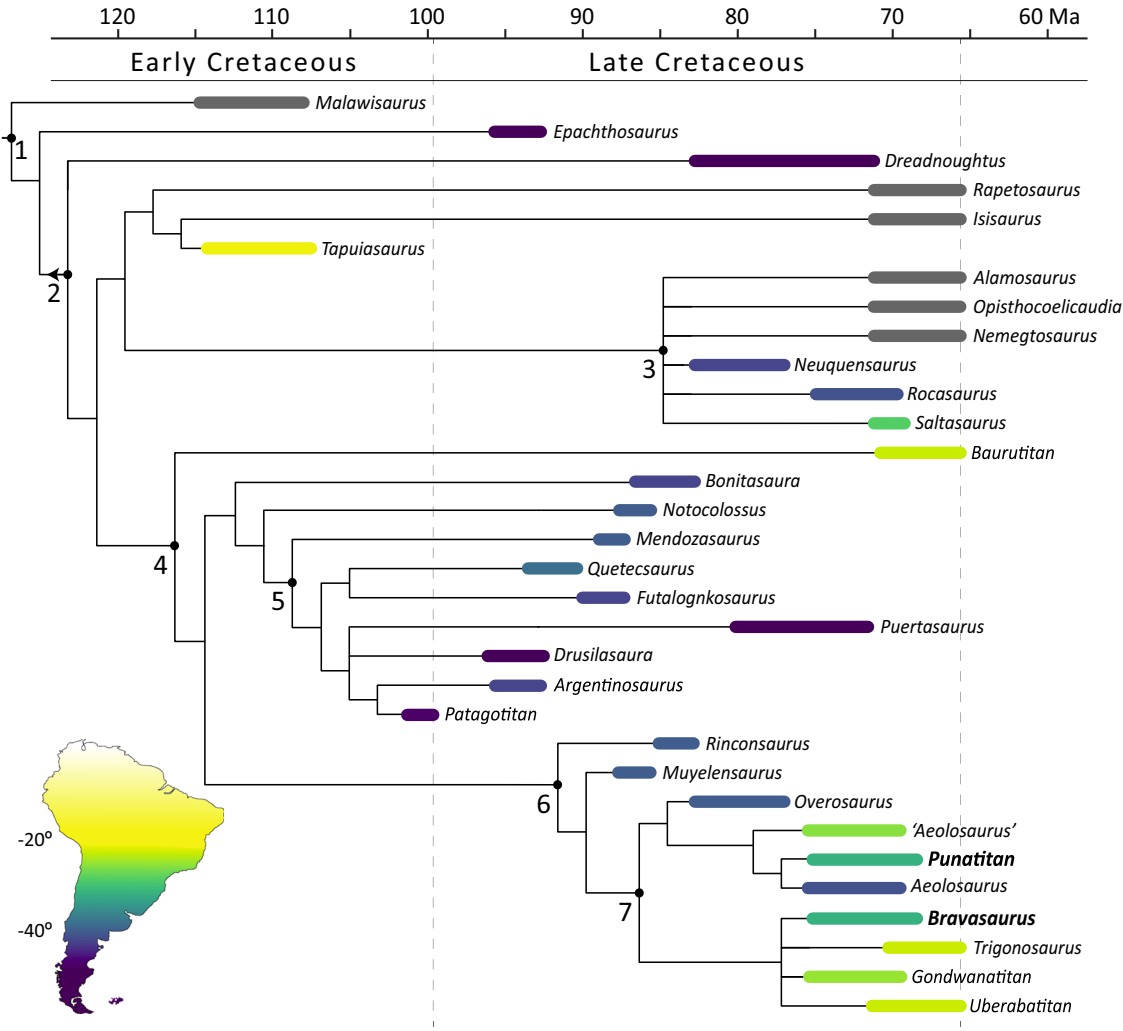

**Fig. 5 Phylogenetic relationships of *Punatitan* and *Bravasaurus* within Lithostrotia.** Phylogeny of derived titanosaurians, based on the data set of Carballido et al. [6] (see "Methods" section and Supplementary Fig. 4). Time ranges for each terminal were obtained from published data. Colours in South American taxa are based on their palaeolatitudinal position. Both time ranges and palaeolatitude are given in Supplementary Table 4. 1. Lithostrotia, 2. Eutitanosauria, 3. Saltasauridae, 4. Colossosauria, 5. Lognkosauria, 6. Rinconsauria, and 7. Aeolosaurini.

noticeable changes in their position (Fig. 5; Supplementary Fig. 4). The former one is placed as the basalmost colossosaurian, and the latter is clustered together with *Uberabatitan*, *Gondwanatitan*, and *Bravasaurus*.

Both *Punatitan* and *Bravasaurus* are recovered within Colossosauria[9]. *Punatitan* shows three of the seven ambiguous synapomorphies that diagnose the newly erected clade[9], and *Bravasaurus* five. Furthermore, the new Riojan species are placed within the clade Rinconsauria, along with several titanosaurians from SW Brazil and Patagonia (Fig. 5). *Punatitan* is nested with the Argentinean *Aeolosaurus*, by sharing the presence of distally expanded prezygapophyses in posteriormost anterior and middle caudal vertebrae. Other features of the caudal vertebrae, such as the dorsal edge of the anterior articular surface of the centrum ahead of the ventral margin, and the neural spines anteriorly oriented in the posteriormost anterior and middle caudal vertebrae, relate the latter taxa with the Brazilian '*Aeolosaurus*' and *Overosaurus*, as successive sister taxa. *Bravasaurus* is included in a collapsed clade comprising the Brazilian *Trigonosaurus*, *Uberabatitan*, and *Gondwanatitan*. The clade is supported by a single synapomorphy: height/width ratio smaller than 0.7 in the posterior articular surface of cervical centra.

**QSD nesting site**. We documented three egg-bearing levels in the lower section of Ciénaga del Río Huaco Formation at QSD. The egg clutches and eggshells are included in an interval of flood-plain deposits in at least three distinct but closely spaced horizons at 59.2, 62.8 and 63.9 m above the base of the unit (Supplementary Fig. 1). Fossil-bearing rocks are siltstones and sandy silt-stones with horizontal lamination and graded and massive bedding that form thin tabular sheets, extending for tens to hundreds of metres. The fossiliferous layer is laterally traced over more than three kilometres, and the egg clutches and eggshells (CRILAR-Pv 620–621) are exposed regularly all along with it. Nineteen egg clutches were spotted, one with up to 15 sub-spherical eggs, arranged in two superposed rows.

The QSD eggs are similar to some Late Cretaceous titanosaurian eggs[69]. Among the remarkable diversity of eggs worldwide, only Auca Mahuevo[70] (Argentina), Dholi Dungri[71] (India), and Toteşti[72] (Romania) preserve titanosaurian embryos. Therefore, these sites are the most reliable to correlate eggs with their producers. At QSD, the eggs are cracked, slightly compressed and flattened by the sedimentary load (Fig. 6a, b). We estimate an egg size of 130–140 mm, similar to the eggs from Auca Mahuevo[70] and Toteşti[72], but slightly smaller than the ones from Dholi

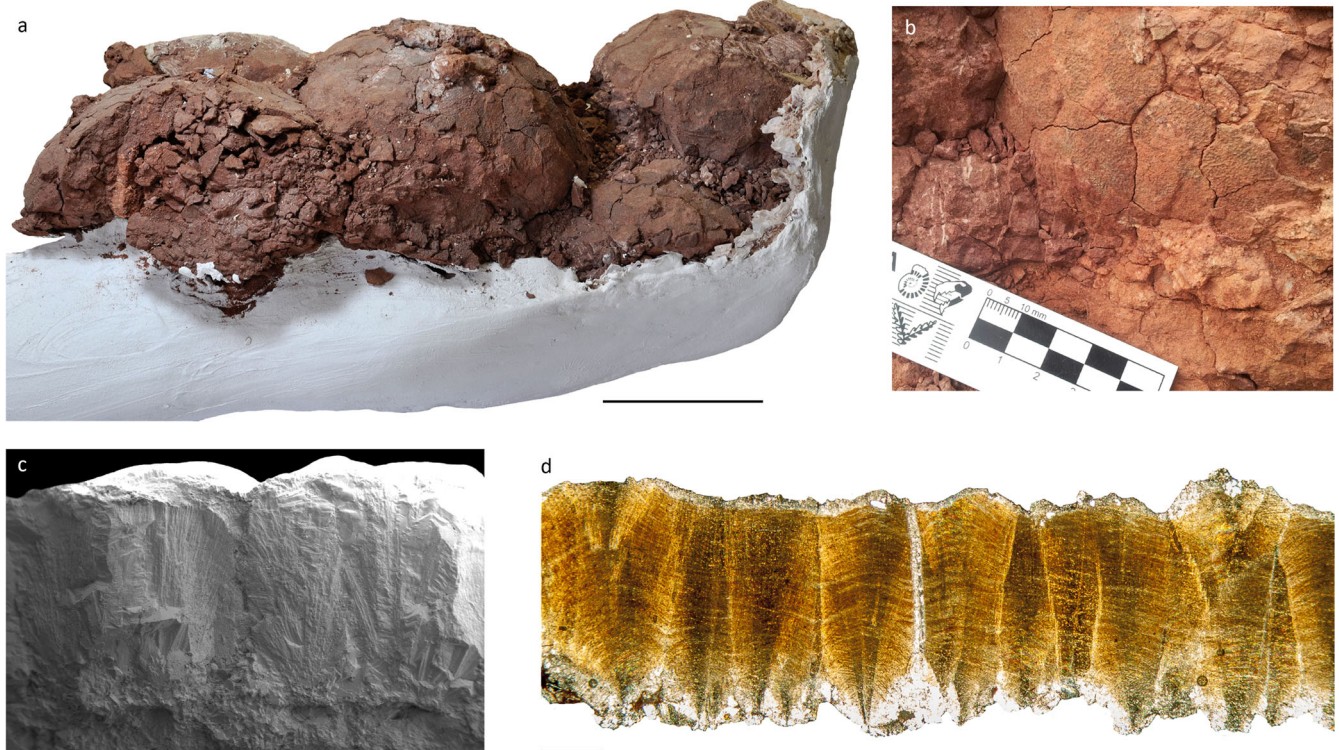

**Fig. 6 Quebrada de Santo Domingo nesting site. a** Part of a titanosaurian egg clutch, CRILAR-PV 620/001. **b** Partial egg and the surrounding matrix. **c, d** Eggshell micrographs under SEM **c** and TLM **d**. Note the straight pore canal with a funnel-shaped aperture in **c**. Scale bars: 100 mm in **a**, and 0.5 mm in **c, d**.

Dungri[71] (160 mm). The eggshells are mono-layered, measuring 1.67 ± 0.31 mm (*n* = 30). The thickness is similar to the eggshells from layers 1–3 of Auca Mahuevo. The eggshells from Toteşti and layer 4 of Auca Mahuevo are slightly thicker, measuring 1.7–1.8 mm, whereas in Dholi Dungri they reach 2.26–2.36 mm. The QSD shells are composed of densely packed shell units of calcite crystals, which radiate from nucleation centres (Fig. 6c, d). They flare out at 50°, and their lateral margins become parallel at the inner third of the shell, like in the Auca Mahuevo specimens[70]. Outwards, the units end out in rounded nodes of 0.3–0.4 mm in diameter, forming densely packed ornamentation that is typical of the titanosaurian clade[69–72]. Multiple straight pore canals run through the eggshell, between the shell units. They have funnel-shaped external apertures that form round depressions between the surficial nodes. Among titanosaurian eggshells, those from Dholi Dungri and Auca Mahuevo (layers 1–3) also have straight pore canals, whereas, in those from Toteşti and the layer 4 of Auca Mahuevo, the pore canals ramify in a Y-shaped pattern.

As in Auca Mahuevo and other Cretaceous nesting sites, the QSD specimens are preserved in a floodplain palaeoenvironment. The occurrence of compact accumulations of whole eggs is consistent with the hypothesis of incubation within the substrate, as currently do the megapode birds from Australasia[69]. Along with the egg clutches, hundreds of shells also appear scattered within the egg-bearing levels. Such an arrangement could be a consequence of the local transport of exposed shells during floods, but also the product of local removal during subsequent nesting episodes. Soft sediment deformation and dislocation are frequent, and could also have contributed to their dispersion. These features suggest that each of the three egg-bearing levels could constitute a time-averaged assemblage.

## Discussion

As far as we know, *Punatitan* and *Bravasaurus* represent the first confirmed occurrence of colossosaurian titanosaurians[9] in NW Argentina. For 40 years, *Saltasaurus* remained as the only well-represented sauropod for this region. *Saltasaurus* is closely related with the Patagonian *Rocasaurus* and *Neuquensaurus*, as well as *Yamanasaurus*[19], from Ecuador. There is a consensus regarding the close relationship of these taxa, which constitute the Saltasaurinae, a clade of small-sized titanosaurians from the Late Cretaceous that is also supported by our phylogenetic result. The phylogenetic data also suggest that saltasaurines may not have a close relationship with other Late Cretaceous titanosaurians from South America (Fig. 5). Fragmentary findings in NW Argentina[20,23,73] and Chile[74] suggested the occurrence of non-saltasaurine titanosaurians between Patagonia and Bauru, but the hitherto known evidence was insufficient to conjecture about their phylogenetic affinities. The new phylogenetic analysis recovers *Punatitan* within a clade of typically "aeolosaurine" taxa, such as *Aeolosaurus* and *Overosaurus*, whereas *Bravasaurus* is nested in a collapsed clade with Brazilian species. The Patagonian and Brazilian *Aeolosaurus* species show a close relationship as previously supported[11], but recent phylogenetic analyses, including the one here presented, suggest the Brazilian species may represent a distinctive genus, other than *Aeolosaurus*[12,13]. Both Riojan species expand the diversity of the clade Rinconsauria, and its geographical distribution.

Based on a combination of direct observations and body mass estimation, *Bravasaurus* was a small-sized titanosaurian, though not as small as the dwarf *Magyarosaurus* or *Lirainosaurus*. Although it had probably reached its maximum size, it is much smaller than *Punatitan* (Fig. 1c, d). The largest titanosaurians ever known are placed within colossosaurians[2,9] (e.g., *Argentinosaurus*, *Patagotitan*), but others are relatively smaller, such as

*Rinconsaurus, Overosaurus, Trigonosaurus, Baurutitan,* and *Gondwanatitan.* In this context, the available evidence suggests that *Bravasaurus* (~3 tons) is the smallest colossosaurian yet recorded, followed by the taxa mentioned above. In contrast to *Magyarosaurus*[68], *Bravasaurus* appears to have inhabited inland territories. By the latest Late Cretaceous, there is an evident reduction in size in saltasaurids and rinconsaurians across South America, which may be related to fluctuations in climate[75] and vegetation[76] (e.g., grassland), as a result of more temperate conditions and influence of remnant epicontinental seas during the dynamic aperture of the Atlantic.

The new findings from La Rioja reduce the paleobiogeographic gap of Late Cretaceous colossosaurians in South America, which were previously restricted to Patagonia and SW Brazil. Colossosauria is divided into the gigantic Lognkosauria (e.g., *Patagotitan, Futalognkosaurus*), plus some related forms, and the Rinconsauria. So far, the former clade is mostly limited to Patagonia (although there are few putative non-rinconsaurians in Brazil[14]), whereas Rinconsauria may contain a few Brazilian forms[2,6,9,77]. Besides, some taxa recovered within Rinconsauria are often included within Aeolosaurini, a group of titanosaurians with unstable interspecific phylogenetic relationships[12]. Our results suggest that Rinconsauria is much more diverse and widely distributed than previously thought[2,3,6,9,37]. The oldest representatives of this clade would be in northern Patagonia, for the earliest Late Cretaceous. By the Campanian–Maastrichtian, the Rinconsauria increased their diversity and spread geographically northward, through La Rioja, to SW Brazil.

Comparison of the QSD eggs with confirmed occurrences of titanosaurian eggs, such as Auca Mahuevo[70] and Toteşti[72], allow their identification. The spherical shape of the eggs, the mono-stratified shells and the nodular external ornamentation indicate that the QSD eggs belong to titanosaurian sauropods. More specific features (e.g., egg size, shell thickness, and straight vertical pore canals), associate the QSD specimens with the Auca Mahuevo eggs (layers 1–3). La Rioja Province is already known for its titanosaurian nesting sites in the Los Llanos region, several hundred kilometres southeast of QSD[78,79]. There, two localities preserve Late Cretaceous nesting sites that show distinct palaeoenvironmental conditions. The eggs from these sites markedly differ in their shell thicknesses but share the same egg diameter, around 170 mm, larger than the 140 mm eggs from QSD. In South America, the only eggs to match that size are those from Auca Mahuevo and Río Negro[80], in Patagonia, as well as an isolated record from Bauru[81]. Eggs similar in diameter were attributed to dwarf Cretaceous titanosaurians from Toteşti[72]. The QSD eggs are relatively small, so either *Bravasaurus* or *Punatitan* may have been the producers. Further specimens are required to evaluate each scenario.

Both the oological and sedimentological data suggest a distinct nesting strategy from other sites of La Rioja. Unlike the sites in Los Llanos, the titanosaurian eggs of QSD appear in successive floodplain deposits, as occurs in Auca Mahuevo and other nesting sites worldwide[69]. Each of the egg-bearing levels contains multiple egg accumulations that were not necessarily laid contemporaneously. The several episodes interspersed in the sedimentary sequence allow us to infer nesting site philopatry, a behaviour that seems to have been frequent among Cretaceous titanosaurians[69,72,78,82,83]. This evidence and egg morphological features advocate a nesting strategy similar to that displayed at Auca Mahuevo. The QSD site provides further evidence on the plasticity of Late Cretaceous titanosaurian sauropods regarding their nesting strategies. Although it is still necessary to better understand the nesting conditions in other regions, such as Brazil, it seems increasingly evident that the adaptation to different nesting strategies could have been crucial in the diversification and dispersal of titanosaurians across South America.

## Methods

**Specimens.** All material described in this study is housed at the Paleovertebrate Collection of CRILAR (La Rioja, Argentina).

**Taxa and systematic definitions.** For the sake of simplicity, we used generic names when they are monotypic. The only exception corresponds to *Aeolosaurus*. The data set already included '*Aeolosaurus*' *maximus*, a taxon which has been recognised as a member of Aeolosaurini[84], although it does not exhibit the diagnostic features of the genus (see Martinelli et al. [12] for further discussion) and is not grouped with the Patagonian species in some analyses[13,14]. Consequently, we refer to it as '*Aeolosaurus*'. We followed the systematic definitions provided by Carballido et al.[2] and González Riga et al.[9].

**Eggshell micro-characterisation.** We selected several eggshell fragments from QSD for microscopic imaging. Thin sections were carried out in the Petrology Lab at CRILAR, La Rioja, using the standard protocol for petrographic sectioning. We cut and mounted six eggshell fragments for their observation under a scanning electron microscope, following the protocol described in a previous study[85]. We used a LEO 1450VP equipment in the Laboratorio de Microscopía Electrónica y Microanálisis (Universidad Nacional de San Luis, San Luis, Argentina).

**Body mass.** We estimated the body mass of *Bravasaurus* using a scaling equation adjusted for phylogenetic correlation/covariance[67]. The equation

$$\log BM = 2.754 \cdot \log C_{H+F} - 1.097$$

where BM is body mass, and $C_{H+F}$ is the sum of circumferences of the humerus and femur. It has been used to estimate the body mass of gigantic (e.g., *Patagotitan*[2]), as well as medium-sized titanosaurians (e.g., *Rapetosaurus*[86]).

**Phylogenetic analysis.** We tested the phylogenetic position of *Bravasaurus* and *Punatitan* amongst 30 derived titanosaurian terminals using a modified version of the data matrix of Carballido et al.[6]. This matrix has been used to assess the phylogenetic position of derived titanosaurians and related taxa (e.g., *Sarmientosaurus*[4], *Patagotitan*[2]).

Data on several South American titanosaurians was added in order to expand the representation of their diversity. We added scorings for *Gondwanatitan* and *Uberabatitan* to increase the information on Brazilian taxa. We also included *Aeolosaurus rionegrinus*[30] and the saltasaurine *Rocasaurus*, from Patagonia to the data set.

We added five characters (four from previous studies and one new) and modified few scorings (Supplementary Tables 4, 5; Supplementary Data 1). This resulted in a data set of 96 taxa and 421 characters (Phylogenetic Analysis in Supplementary Information, and Supplementary Data 2). As in previous studies[6], 24 characters were considered as ordered (14, 61, 100, 102, 109, 115, 127, 132, 135, 136, 167, 180, 196, 257, 260, 277, 278, 279, 280, 300, 304, 347, 353, 355).

**Statistics and reproducibility.** We performed a parsimony analysis of the modified data matrix using TNT v.1.1[87]. We did a heuristic search with 1000 replicates of Wagner trees and two rounds of tree bisection-reconnection branch swapping. Branch support was quantified using decay indices (Bremer support values). They were calculated with TNT v.1.1[87], and are given in the Supplementary Fig. 4. A TNT file containing raw data for the parsimony analysis is available in the Supplementary Data 2.

**Nomenclatural acts.** This published work and the nomenclatural acts it contains have been registered in ZooBank, the proposed online registration system for the International Code of Zoological Nomenclature (ICZN). The ZooBank LSIDs (Life Science Identifiers) can be resolved and the associated information viewed through any standard web browser by appending the LSID to the prefix "http://zoobank.org/". The LSIDs for this publication are: urn:lsid:zoobank.org:pub:CDA87D24-50DA-415A-9FAF-54FB7CF26D73; urn:lsid:zoobank.org:act:18840DCF-33EF-465D-8F69-0B38BB601BF7; urn:lsid:zoobank.org:act:658B5D64-1432-46BC-B543-DFF1155EC71E; urn:lsid:zoobank.org:act:336215DA-56AB-4B69-8059-C1FFA564D58A; urn:lsid:zoobank.org:act:84B7ECE6-60B4-4324-B983-CD86C8952E8A.

**Reporting summary.** Further information on research design and fieldwork is available in the Nature Research Reporting Summary linked to this article.

## Data availability

Additional information, including the dataset analysed in this study, is available in the Supplementary Information, and Supplementary Data 1, 2 files. CRILAR-Pv 612-614 and

620-621 are deposited at the Paleovertebrate Collection of CRILAR (Anillaco, La Rioja), and are available upon request.

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

## Acknowledgements

We thank Secretaría de Cultura, Subsecretaría de Patrimonio, and Gobierno de La Rioja. We are grateful to Sergio de la Vega, Carlos Bustamante, Julia Desojo, Hernán Aciar, Leonel Acosta, Marcelo Miñana, Victoria Fernandez Blanco, Tomaz Melo, Jimena Trotteyn, Mariano Larrovere, Marcos Macchioli, Tatiana Sánchez, Gabriel Hechenleitner, Eugenio Sanchez, and Walter Bustamante for their help during fieldwork and preparation of the specimens. Special thanks to Grupo Roggio, and Alberto Acevedo, Germán Brizuela, Rubén Brizuela, and Carlos Verazai, for providing us with accommodation at their facilities during fieldworks (2016–2019). We thank Tim Coughlin, Rod Holcombe, and Andrea Arcucci for sharing information about QSD. Comparative data were collected in previous visits thanks to Pablo Ortiz (Instituto Miguel Lillo, Tucumán), Ignacio Cerda (Museo Carlos Ameghino, Río Negro), Diógenes de Almeida Campos and Rodrigo Machado (MCT), and Luiz Carlos Borges Ribeiro (CPPLIP-UFTM). We thank Jose Carballido for his help with the phylogenetic analysis. The Willi Henning Society for TNT software. Research supported by CONICET, The Jurassic Foundation (E.M.H.), and the Palaeontological Society PalSIRP (E.M.H.). Fieldwork permits (2015–2019): RES S.C. 135, 238, 109, 124.

## Author contributions

E.M.H. coordinated the project (including fieldwork); E.M.H., A.G.M., L.E.F. were involved in study concept. E.M.H., L.L., A.G.M., and S.R. wrote the paper, and with J.R.A.T., L.E.F., and L.S. gathered the data. L.L., L.E.F., and S.R. made the figures. All authors discussed the results and commented on the manuscript.

## Competing interests

The authors declare no competing interests.
