## [Peer Review File · Communications Biology]

Reviewers' comments:

Reviewer #1 (Remarks to the Author):

This is an interesting and mostly well-written paper about two new titanosaurian dinosaur species and a new titanosaurian nesting site. The two new taxa, respectively, cluster phylogenetically with Patagonian and Brazilian titanosaurs: thus, the new taxa provide a link between the two otherwise disparate faunas. Incorporation of data pertaining to the new titanosaur *Yamanasaurus* from Ecuador, which also shows ties to Patagonian taxa, will be essential, but otherwise this is up-to-date.

What I would have liked to see, given that it is repeatedly highlighted, is some speculation as to why *Bravasaurus* (and saltasaurines generally) are so small (relatively speaking). The possible reason is even alluded to in the Introduction - inundation of South America by inland seas, which could have divided the continent into several disparate large islands. Perhaps the western one, where saltasaurines held sway, was smaller than the eastern one (Brazil), where sauropods might have stayed larger. Also, the size of *Bravasaurus* should be compared against other small sauropods (as it is in the Abstract), rather than medium-sized forms.

This paper will be of particular interest to titanosaurian workers in South America, but will also be of global interest among sauropod workers.

I recommend minor revisions and look forward to seeing this published.

Stephen Poropat

Reviewer #2 (Remarks to the Author):

The study of Hechenleitner and co-authors represent an important contribution to the knowledge of the taxonomic diversity and geographic distribution of South American titanosaur sauropods. The study describes two new species of titanosaurians that fill the geographical gap within the Colossosauria clade, providing evidences of a more widespread distribution than previously though. In addition, the authors report the discovery of a new titanosaurian nesting-site comparable in size to the renowned Auca Mahuevo (Argentina).

Overall, the manuscript is well written and has a coherent structure. Main goals of the study are set clear. Despite the requirements of the journal regarding the extension of the text, descriptions and comparisons are enough to support the singularity of the new taxa. The methodology for the phylogenetic exploration is appropriate. The topic selected for the study is of broad interest, not only for specialist in titanosaurian sauropods, but also for any scientist that is interested in the study of distribution and dispersal factors of extinct terrestrial animals. Nevertheless, there are some aspects that must be addressed before considering the acceptance of the manuscript for publishing.

- 1) The introduction must be improved in order to give a more precise idea of the idiosyncrasy of the Colossosauria clade.
- 2) The description and comparison of QSD eggshells needs a review, and it should be related with the main goal of the study. Otherwise it should be considered to exclude from the study.
- 3) The time calibration of the phylogenetic study, especially the establishment of time-divergence of the clades, is unclear.
- 4) Discussion fell short in some aspects (see below).

SPECIFIC COMMENTS:

1/ I do not pretend to be fussy, but (like some reviewers already did to me) I have a general observation in the usage of the term "titanosaur" or "titanosaurs".

Despite being widely (commonly and colloquially) used, the term titanosaur – referring to members of the clade Titanosauria – is incorrectly used, and should be avoided. Reasons for such proposal are based on the fact that the word "titanosaur" derives from the genus Titanosaurus, so it makes reference *sensu stricto* to an individual of this specific genus. However, Wilson and Upchurch (2003) considered Titanosaurus as *nomen dubium*, and consequently the derived term also lost its validity.

In this regard, according to the ICZN, the most similar and valid nomenclatural term is: Titanosauria (Bonaparte and Coria, 1993), and as such the derived term should be titanosaurian or titanosaurians.

INTRODUCTION

2/ The introduction of the manuscript must be improved. Now, it provides a general view of the distribution of titanosaurs in South America, and pay special attention to the absence of saltatorids in Brazil; but nothing is mentioned about the clade that the authors focus on: Colossosauria. Some lines explaining the distribution, abundance or relationship with saltatorids would be appreciated.

3/ Although it can be deduced, there is no direct reference of the institution where the Punatitan and Bravasaurus type material is housed. This observation leads me to realize that there is no specific "Institutional abbreviation" section, so the acronym CRILAR is only resolved in the authors affiliation. Therefore, I suggest including a line solving all these questions:

All material described in this study is housed in the collections of Centro Regional de Investigaciones Científicas y Transferencia Tecnológica de La Rioja (CRILAR; La Rioja, Argentina)

4/ However, although its significance, I do not see the purpose to include the report of the new nesting site given that it seems to have no special relation with the main goal of the study. If authors want to keep this part, they should find a way to incorporate the occurrence of new egg remains to the general discussion of titanosaur geographic distribution or, if any prove, to link the eggs to any of the two new species.

I added some specific comments regarding this section in the comments for the authors.

5/ Lines 46-47: authors mentioned that both pleurodiran turtle and notosuchian crocodylomorphs exhibit similar heterogeneous distribution than titanosaurs, but other group of dinosaurs (which could exhibit similar dispersal capabilities) too, like abelisaurid theropods (Delcour, 2018). I think that it would be appropriate, at least, to include some mention to any taxonomic group that has more close phylogenetic relationship to sauropods.

R. Delcour. 2018. Ceratosaur palaeobiology: new insights on evolution and ecology of the southern rulers. *Scientific Reports* 8: 9730.

6/ Lines 53-55: Despite there is no evidence of saltatorids in Brazil; I wonder how the recent discovery of Yamanasaurus (Apesteguía et al., 2020) from Ecuador can influence the results of the study. It would be appreciated to make some mention to this question, at least, in the discussion.

Apesteguía, S., Soto Luzuriaga, J.E., Gallina, P.A., Tamay Granda, J., & Guamán Jaramillo, G.A. (2020). The first dinosaur remains from the Cretaceous of Ecuador. *Cretaceous Research* 108: doi:10.1016/j.cretres.2019.104345

RESULTS

Systematic paleontology

Systematic descriptions are appropriated and arguments to erect new taxa are convincing. Since I'm not an expert in titanosaurian taxonomy I have very little to say in this section:

7/ Line 75: If authors want to be consistent along the text and use American English, then they should use paleontology instead of palaeontology.

8/ Line 144: The figure 2 caption indicates that "circled number correspond to synapomorphies in the text". However, the number 1 is marked in Fig2a, which corresponds to a cervical vertebra (C12), while the first synapomorphy, according to the text, is referred to middle dorsal vertebra (D6-D7) as correctly indicated in Fig2d. The miss-indication on figure should be emended.

9/ Line 223: "centra is spongy" change for "cantra are spongy"

10/ Lines 295-302: The division of the articular condyle of the quadrate is a very peculiar character indeed. I do not doubt of the validity of this character, but it is difficult to tell just from a picture; could you provide a better (perhaps more contrasted) picture? Can you ensure that this feature is not produced by taphonomic means?

When comparing this character, what is the condition of the articular process in South American titanosaurs Tapuisaurs and Sarmientosaurus?

11/ Lines 302-307: I am unable to recognize the quadratojugal in Fig.3a.

Is it like I tried to indicate in the figure?

Even if the quadratojugal extends more backward, how can the authors infer the projection of the anterior process of the bone? This region is not preserved in the figured specimen.

12/ Lines 374-375: In addition to the current comparison, the deltopectoral crest is also markedly expanded distally in *Dreadnoughtus* and *Opisthocoelicaudia*, just if authors want to include this information to the description.

13/ According to Fig. 1 both *Punatitan* and *Bravasaurus* preserve rib elements. One significant synapomorphy among titanosaurians is the occurrence of pneumatic foramen at the proximal end of the dorsal rib (see Mocho et al 2019). Is this feature present in the new taxa?

Mocho, P., Pérez-García, A., Martín Jiménez, M., & Ortega, F. (2018). New remains from the Spanish Cenomanian shed light on the Gondwanan origin of European Early Cretaceous titanosaurs. *Cretaceous Research*. doi:10.1016/j.cretres.2018.09.016

14/ Lines 389-394: Assuming that BM estimation is correct, 2.9 tones is a really low value for a titanosaur. The purported "dwarfs" titanosaurians *Ampelosaurus* from France and *Lirainosaurus* from Spain are estimated to be 2.5 and 1.8 tones respectively (see Sup. Excel file of Bensot et al. 2012). I realize that it is not the main scope of the paper, but I think that, apart of solely mentioning that *Bravasaurus* is the smallest colossosaurian yet recovered (lines 479-480), authors should propose some hypothetical scenario that helps to explain the occurrence of such small "titano" among colossal beast.

Phylogenetic analysis and biogeographic distribution

The phylogenetic results are convincing and help to depict a remarkable paleobiogeographic pattern. It is noteworthy the huge effort that the author made in re-coding several taxa, which sure help improved the final results. However, I have some comments for this section:

15/ Although Bremer support index is provided in the Supplementary Material, I miss the report of other important phylogenetic indices, such as RI and CI. This information is useful to give an idea of the "robustness" of the results.

16/ Fig.4 I recommend to somehow highlighting the position of both Punatitan and Bravasaurus in the phylogeny, just to make visually easier its location.

17/ Since TNT is unable to establish time calibrations, to me, it remains unclear how authors established the time of divergence (nodes) between groups/taxa in the phylogeny. Did you consider any specific ratio of speciation/evolution? Did you place them according previous studies? This part should be improved given that it should affect parts of the discussion (Lines 481-493).

18/ I have to admit that I am not familiar with the paleogeographic characteristics of South America at the end of the Cretaceous (Campanian-Maastrichtian), and therefore I have some questions about this topic and how author established the criteria to conduct the paleobiogeographic analyses.

19/ If I am right, authors establish a "current" latitudinal gradient in order to define the location of different titanosaure taxa. But, does the geographic position (latitude/longitude) of South America have remained invariable for the last 80-66 Ma?

If the answer is yes (or almost negligible) then I agree that the current location of the fossil sites can be assumed as the same as that from 70 Ma. Otherwise, if the South American continent experienced any significant displacement on the lithosphere during the last millions of years, then the geographic position should be corrected and the paleogeography analysis re-run considering past latitude/longitude locations.

QSD Nesting Site

20/ Despite its remarkable significance, it seems that this section is somehow off place and disconnected from the main purpose of the study, which – as far as I could understand – seems to be focused on provide evidence of the wide geographic distribution of the clade Colossosauria.

21/ However, since dinosaur eggs and reproduction is my main topic of study, I have some comments for the authors about this section.

22/ By contrasting the information provided in both the main text and SI, I found some problems to set a clear picture of the occurrence of titanosaure eggs in terms of geographic and stratigraphic distribution. Following I list the three main issues that I found:

1) Stratigraphic distribution

23/ The distribution of the 3 egg-bearing strata needs some clarification. In the main text it is mentioned that they outcrop within a 30 m thick sequence, which I suspect that it refers to the "grey interval" compressed between 45 and 70-75m above the base of Ciénaga del Río Hueco Fm. (according to Fig S1b), although no indication is provided in this regard. On the other hand, in the Supplementary Information it is said that they are distributed in a 5 m thick interval, that according to the text it is located 50-65 m above the base of the formation. First, that is actually 15 m interval not 5m, and second the interval is located around 60-65m from the base according to Fig. S1b.

So, what is the real stratigraphic distribution of the 3 egg-bearing levels? Please, improve this part.

24/ In lines 429-430 it is stated that eggs and eggshells occur at different (stratigraphic?)

positions within the floodplain depositional sequence and, therefore, it is likely to be interpreted that they represent several nesting events.

However, when the authors say "several nesting events", would they like to mean "three", as the number of strata? How can the authors establish the number of nesting events?

Additionally, could it be that eggs with different positions within the profile could represent eggs of the same clutch but in a distinct internal location (in line 441 it is mentioned that eggs can occur in two-row arrangement) rather than different events?

2) Lateral extension of the nesting site

25/ The way in which the information is provided in the main text (lines 425 to 427) seems to suggest that the accumulations of eggs can be traced laterally over three kilometers, which would imply a humongous nesting area.

However, information from Supplementary Information seems to point that it is the stratigraphic interval (lithology/strata) the one that can be traced for more than 3 km, which leads to a total different interpretation.

Thus, I suggest clarifying this point: Where egg remains were recovered along 3 km? Or were they found in a single site (spot)?

26/ In addition to all of the above, it would be useful to provide the number of clutches, eggs, and eggshells discovered in the new nesting site, just to give a clear idea of the abundance and richness of the locality.

3) Time occurrence

27/ Authors seem to be somehow conditioned for the Auca Mahuevo nesting site, and omit many other evidences of "nesting site fidelity" in titanosaurs. Additional examples of nesting recurrence in a single site can be found in both South America (see Salgado et al., 2007, 2009) and Europe (Sellés et al., 2013, 2017). I recommend including those additional references.

Salgado L, Coria RA, Ribeiro CM, Garrido A, Rogers R, Simón ME, Arcucci AB, Rogers KC, Carabajal AP, Apesteguía A, et al. 2007. Upper Cretaceous dinosaur nesting sites of Río Negro (Salitral Ojo de Agua and Salinas de Trapalcó-Salitral de Santa Rosa), northern Patagonia, Argentina. *Cretaceous Res.* 28:392–404.

Salgado S, Ribeiro CM, García RA, Fernández MS. 2009. Late Cretaceous Megaloolithid eggs from Salitral de Santa Rosa (Río Negro, Patagonia, Argentina): inferences on the titanosaurian reproductive biology. *Ameghiniana.* 46:605–620.

Sellés AG, Bravo M, Delclòs X, Colombo F, Martí X, Ortega-Blanco J, Parellada C, Galobart A. 2013. Dinosaur eggs in the Upper Cretaceous of the Coll de Nargó area, Lleida Province, south-central Pyrenees, Spain: oodiversity, biostratigraphy and their implications. *Cretaceous Res.* 40:10–20.

Sellés AG, Vila B, Galobart, 2017. Evidence of Reproductive Stress in Titanosaurian Sauropods Triggered by an Increase in Ecological Competition. *Scientific Reports* volume 7, Article number: 13827.

28/ Line 432-433: It seems that "Auca Mahuevo" is repeated twice within the same phrase. "as reported for Auca Mahuevo the world-famous Auca Mahuevo nesting site, in the Argentine Patagonia"

It should be: "as reported for the world-famous Auca Mahuevo nesting site, in the Argentine Patagonia"

29/ Line 440: Authors starts the identification of QSD eggs mentioning that they share many features with those discovered in the Auca Mahuevo nesting site. But in my opinion the description of the QSD eggshell is so general that such features also agrees with several egg-types from India, Spain, France, Romania, Brazil, and several site from Argentina (i. ex. Salitral Rosa, Salitral Oje de Agua, Salitral Moreno, or Neuquen City).

Thus, I suggest developing this section a little more. Apart from the shell thickness, shell unit shape, and pore canal system, authors should focus the new description and discussion on two additional features: 1) shape and contact between adjacent shell units, and 2) H:W ratio of the shell units.

1) shape and contact between adjacent shell units.

According to the most recent reviews on megaloolithid eggshells (see Fernández and Koshla, 2017 for further details) the way in which adjacent shell units growth allows to distinguish to main groups of megaloolithid eggs: Megaloolithus-with well-defined and straight edges; and Fusioolithus-with fused edged.

Eggs from Auca Mauevo have been included in the later group (Fernández and Koshla, 2017), while those from QSD site seems to posse well-defined shell margins (personal interpretation based on Fig.5d). I recommend the authors to take a look at the samples from the studies of Salgado et al (2007).

2) H:W ratio of the shell units.

This feature is commonly used to discriminate between "oospecies" of megaloolithid eggs (see Fernández and Koshla, 2017).

It seems that in QSD shells this values is about up to 2.5:1 or so, which is much closer to the specimens from Neuquen City (*Megaloolithus jabalparuenis*) than that of Auca Mahuevo (*Fusioolithus bahensis*).

Fernández Mariela S. & Khosla Ashu (2014): Parataxonomic review of the Upper Cretaceous dinosaur eggshells belonging to the oofamily Megaloolithidae from India and Argentina, *Historical Biology: An International Journal of Paleobiology*, DOI: 10.1080/08912963.2013.871718

I believe that, by following my recommendations and processing a little bit the oological information, author should be able to extract some additional paleobiogeographic conclusions that may help to support their hypotheses.

Moving to the discussion section reserved for QSD eggs (starting from Line 494), this part must be largely improved.

30/ Line 496: As aforementioned, the features exhibited by QSD eggs are common for several sites from South America (Peru, Brazil, Argentina, Uruguay), Europe (Spain, France), India, and even Africa (Morocco, Tanzania); so to restrict the comparison to Auca Mauevo and Totesti seems too poor.

31/ Line 497: I do not see the porpoise to compare the eggs from QSD to those from Los Llanos region, especially because they belong to a different type of sauropod eggs. Those from La Rioja belong to the group of Faveoololithus (extremely thick eggshells with complex reticulate pore system), while it seems that those from QSD are attributable to Megaloolithus (more thin eggshell with simple pore system).

32/ Lines 500-504: The size of the egg is not a valid criterion for comparison. Eggs of the same eggshell-type can greatly vary in size (see Vianey-Liaud et al., 2003, Fernández and Khosla, 2014). Although the size of eggs from both QSD and Totesti can be similar, this is not enough to use this criterion to support the small body size of new taxa. Eggs of 14 cm in diameter are also reported from Auca Meuevo and several localities from India, and coeval titanosaurian taxa are mid- to large-size.

33/ Line 514: Please, add the bibliographic reference Selles et al (2013), in which is described the longest nesting site recurrence, with up to 40 stratigraphic levels with nests that correspond to some millions of years in time.

Sellés et al., 2013. Dinosaur eggs in the Upper Cretaceous of the Coll de Nargó area, Lleida Province, south-central Pyrenees, Spain: Oodiversity, biostratigraphy and their implications. *Cretaceous Research* 40: 10-20

SUPPLEMENTARY INFORMATION

34/ Line 236: there, it is referred a "Figure X", which I suspect that "X" means to be "S4". Please, change this small error.

35/ Page 17: Perhaps this is because we are reviewing an optimized version of the manuscript, but the quality of Fig. S4 is very low. Please, ensure that this image is in good quality in the final version.

Reviewer #3 (Remarks to the Author):

The paper is interesting and the materials are important. All comments are made via track changes in the attached document. I combined the supplemental info and main text into one document. More justification needs to be present in the main text; as it stands, so much is relegated to the supplemental information, the main paper doesn't stand on its own. The phylogenetic analysis unfortunately doesn't appear to have been very carefully done. I did not go through each and every character (let alone each character scoring), but looking over the list even briefly, I found a duplicate (copied and pasted) character, typos, and potential overlap among characters put forth as disparate. These issues will not give the reader confidence in the results of the phylogenetic analysis, if it is left in. I don't think a phylogenetic analysis is necessary, you could just note synapomorphies present in the material based on previous analyses. This would free up room for describing the provenance, age, and taphonomy of the specimens, all of which are severely underreported in the main text (but present in nice detail in the supplemental information). In sum, I think these are very important fossils and congratulate the authors for the huge undertaking involved in bringing them to light, but I feel that there are substantial issues that prevent publication of the manuscript in its present form.

Replies to Reviewer #1

GENERAL COMMENT

What I would have liked to see, given that it is repeatedly highlighted, is some speculation as to why *Bravasaurus* (and saltasaurines generally) are so small (relatively speaking). The possible reason is even alluded to in the Introduction - inundation of South America by inland seas, which could have divided the continent into several disparate large islands. Perhaps the western one, where saltasaurines held sway, was smaller than the eastern one (Brazil), where sauropods might have stayed larger. Also, the size of *Bravasaurus* should be compared against other small sauropods (as it is in the Abstract), rather than medium-sized forms.

Reply: We added more comparisons (including dwarf sauropods from Europe) and we also provide a potential explanation for the relatively small size among the latest Late Cretaceous titanosaurs from South America.

INTRODUCTION

Lines 41 and 44 These are rather old references.

Reply: We added most of the suggested references that improved the manuscript.

Line 53 *Yamanasaurus* now as well – cited in the supplementary information but not here, for some reason.

Reply: We added *Yamanasaurus* plus the respective reference.

RESULTS

Line 116 The centrum is depressed, with...

R#1: In what way? Do you mean it is shorter dorsoventrally than it is wide transversely?

Reply: We modified the text accordingly. Now it is:

The centrum is shorter dorsoventrally than it is wide transversely, with...

Line 129 spinoprezygapophyseal fossa (sprf).

R#1: Following Wilson et al. (2011), this is the sprf, not the sprlf.

Reply: Corrected.

Lines 212-213 As in all titanosaurs, these caudal vertebrae have strongly procoelous centra¹

R#1: Nope. For example: *Opisthocoelicaudia*: opisthocoelous; *Savannasaurus*: amphicoelous; *Andesaurus*: very weakly procoelous.

Reply: The reviewer is right. We slightly modified the sentence:

As in most titanosaurs, these caudal vertebrae have strongly procoelous centra (Salgado *et al.*, 1997).

Lines 236-238 The haemal arches are similar to those reported for other derived titanosaurs.

R#1: So...? I presume they're open dorsally, with blades that are not expanded anteriorly or posteriorly. But how can I know? Perhaps describe at least a little.

Reply: The haemal arches are not the most diagnostic bones and we have a word limit for this submission. We added some information about them:

The available haemal arches are opened Y-shaped, with no expanded pedicels, as those reported for other derived titanosaurs (Otero *et al.*, 2011).

Lines 267-271_The holotype of *Bravasaurus* (Fig. 3), as well as the referred specimen, indicates a small-sized titanosaurian, much smaller than *Punatitan* (Fig. 1c,d) and other medium-sized sauropods, such as *Trigonosaurus*, *Overosaurus*, and *Bonitasaura*.

R#1: Given this, maybe compare and contrast with small titanosaurs? *Saltasaurus*, *Neuquensaurus*, *Rocasaurus*, *Magyarosaurus*, *Europasaurus*?

Reply: We agree with the reviewer. We added the following sentence:

Considering that both specimens could be adults (see below), they would be similar to *Neuquensaurus* or *Magyarosaurus*.

Lines 271-273_Cranial elements include partial right quadrate and quadratojugal (Fig. 3a,b). The quadrate is anteroventrally directed and bears part of the quadrate fossa.

R#1: It is hard to make much of either of these elements from the figure. Please annotate and/or provide more views. Maybe dash in the outline of the complete element.

Reply: We modified the Fig. 3. Now the lateral and ventral views are enlarged and we added an interpretative drawing for the lateral view and more labelling.

Lines 333-334_The preserved anterior and middle caudal vertebrae of *Bravasaurus* are procoelous as in Titanosauria¹.

R#1: See exceptions above.

Reply: We deleted “as in Titanosauria”, as it is already stated that procoelous caudals are frequent in this clade in lines 234-235. Now it reads:

The preserved anterior and middle caudal vertebrae of *Bravasaurus* are procoelous.

Lines 337-338_The neural arches are on the anterior portion of the centra, as in most titanosaurians.

R#1: This character is more widespread within Sauropoda – for example, *Wintonotitan* shows it.

Reply: We totally agree with the reviewer. However, as space is limited, we restricted the comparisons to the titanosaurian in-group. We slightly modified the sentence:

The neural arches are on the anterior portion of the centra, as in most titanosaurians, and some other titanosauriforms (e.g. *Wintonotitan*).

Lines 351-353_Its robustness is high (RI(Wilson and Upchurch, 2003) = 0.35), as in *Opisthocoelicaudia* (Borsuk-Białynicka, 1977), much more than in *Rinconosaurus* and *Muyelensaurus*.

R#1: Also *Diamantinasaurus* and *Savannasaurus*.

Reply: Modified.

Its robustness is high (RI(Wilson and Upchurch, 2003) =0.35), as in *Opisthocoelicaudia* (Borsuk-Bialynicka, 1977), *Diamantinasaurus* (Poropat *et al.*, 2015), and *Savannasaurus* (Poropat *et al.*, 2016), much more than in *Rinconsaurus* and *Muyelensaurus*.

Lines 363-364...whereas in *Rinconsaurus* and *Muyelensaurus* is located in the middle third.

R#1: Also *Diamantinasaurus*.

Reply: Done.

...whereas in *Rinconsaurus*, *Muyelensaurus*, and *Diamantinasaurus* (Poropat *et al.*, 2015) is located in the middle third.

DISCUSSION

Lines 447-449 *Punatitan* and *Bravasaurus* represent the first confirmed occurrence of colossosaurian titanosaurs⁶⁵ in NW Argentina. For 40 years, *Saltasaurus* remained as the only well represented sauropod taxon for this region.

R#1: True. In the scheme of things, however, *Atacamatitan* is not from that far away... ditto *Yamanasaurus*.

Reply: Yes indeed, *Atacamatitan* is not far from NW Argentina but it is not conclusive that it is a colossosaurian. In lines 458-461, we already provided more information regarding this Chilean species and other fragmentary findings in NW Argentina, which, as best, can be referred as non-saltosaurine titanosaurs. In addition, *Yamanasaurus* is considered a saltosaurine, and its record is far away from NW Argentina. Distance between Ecuador and NW Argentina (c. 3000 km) almost doubles the distance between the latter and Patagonia (c. 1500 km).

FIGURES

Fig. 2_a, Cervical vertebra (C12) in dorsal view.

R#1: Would be good to see this in other views too.

Reply: We agree with R#1. We modified the figure accordingly, by adding a picture of the cervical vertebra in anterior view (Fig. 2a).

Fig. 2.

SUPPLEMENTARY INFORMATION

Lines 38-47_These ages are much older than Maastrichtian... is that not problematic?

Reply: The age of Ciénaga del Río Huaco is controversial. The outcrops at QSD seem to correlate with the uppermost sections of southern exposures in La Rioja and San Juan provinces. We modified the text between lines 38 and 47, to clarify on this.

Supplementary Table 3_Maybe order each region's taxa by age? And/or provide more details about them? How old are they? Specimen numbers and constituents? Horizon, locality? References?

Also, in our paper on *Austrosaurus mckillipi* (Poropat et al. [2017] Alcheringa), we made a case for *Triunfosaurus* not being a titanosaur.

Reply: We fixed the error regarding *Triunfosaurus*. We provided information about age and basins for all lithostrotians (advanced titanosaurians) from South America in the Supplementary Table 4. The list of taxa in Supplementary Table 3 is just raw data for Fig. 1a.

Line 220_This is the type species – why is it in quotation marks? Maximus and colhuehuapensis are the referred species.

Reply: Yep, typing error. Corrected.

Reviewer #2

SPECIFIC COMMENTS:

1/ I do not pretend to be fussy, but (like some reviewers already did to me) I have a general observation in the usage of the term “titanosaur” or “titanosaurs”. Despite being widely (commonly and colloquially) used, the term titanosaur – referring to members of the clade Titanosauria – is incorrectly used, and should be avoided. Reasons for such proposal are based on the fact that the word “titanosaur” derives from the genus *Titanosaurus*, so it makes reference sensu stricto to an individual of this specific genus. However, Wilson and Upchurch (2003) considered *Titanosaurus* as nomen dubium, and consequently the derived term also lost its validity. In this regard, according to the ICZN, the most similar and valid nomenclatural term is: Titanosauria (Bonaparte and Coria, 1993), and as such the derived term should be titanosaurian or titanosaurians.

Reply: We addressed this issue in the manuscript.

INTRODUCTION

2/ The introduction of the manuscript must be improved. Now, it provides a general view of the distribution of titanosaurs in South America, and pay special attention to the absence of saltatorids in Brazil; but nothing is mentioned about the clade that the authors focus on: Colossosauria. Some lines explaining the distribution, abundance or relationship with saltatorids would be appreciated.

Reply: We introduced three phrases in order to provide information about the clade Colossosauria (lines 58-64).

3/ Although it can be deduced, there is no direct reference of the institution where the *Punatitan* and *Bravasaurus* type material is housed. This observation leads me to realize that there is no specific “Institutional abbreviation” section, so the acronym CRILAR is only resolved in the authors affiliation. Therefore, I suggest including a line solving all these questions:

All material described in this study is housed in the collections of Centro Regional de Investigaciones Científicas y Transferencia Tecnológica de La Rioja (CRILAR; La Rioja, Argentina).

Reply: We clarified this by adding “Paleovertebrate Collection of Centro Regional de Investigaciones Científicas y Transferencia Tecnológica de La Rioja, La Rioja, Argentina” next to the first reference of CRILAR-Pv (lines 85-87). Also added a sentence in Methods section (lines 523-524).

4/ However, although it is significant, I do not see the purpose to include the report of the new nesting site given that it seems to have no special relation with the main goal of the study. If authors want to keep this part, they should find a way to incorporate the occurrence of new egg remains to the general discussion of titanosaur geographic distribution or, if any prove, to link the eggs to any of the two new species.

I added some specific comments regarding this section in the comments for the authors.

Reply: Yes, we want to keep this part and some sections of the manuscript were modified to address this issue.

5/ Lines 46-47: authors mentioned that both pleurodiran turtle and notosuchian crocodylomorphs exhibit similar heterogeneous distribution than titanosaurs, but other group of dinosaurs (which could exhibit similar dispersal capabilities) too, like abelisaurid theropods (Delcour, 2018). I think that it would be appropriate, at least, to include some mention to any taxonomic group that has more close phylogenetic relationship to sauropods.

R. Delcour. 2018. Ceratosaur palaeobiology: new insights on evolution and ecology of the southern rulers. *Scientific Reports* 8: 9730.

Reply: In our view, there is no evidence to say that the abelisaur fossil record is taxonomically distinctive between Brazil and Argentina, mostly because the Brazilian records are very limited. But what is known shows certain homogeneity. In contrast, turtles show more differences: i) chelids are not recovered in Brazil; and ii) notosuchians also show different patterns, baurusuchids are much diverse in Brazil and sphagesaurids are unknown in Argentina.

6/ Lines 53-55: Despite there is no evidence of saltasaurids in Brazil; I wonder how the recent discovery of *Yamanasaurus* (Apesteguía et al., 2020) from Ecuador can influence the results of the study. It would be appreciate to make some mention to this question, at least, in the discussion.

Apesteguía, S., Soto Luzuriaga, J.E., Gallina, P.A., Tamay Granda, J., & Guamán Jaramillo, G.A. (2020). The first dinosaur remains from the Cretaceous of Ecuador. *Cretaceous Research* 108: doi:10.1016/j.cretres.2019.104345

Reply: The recent discovery of *Yamanasaurus* has been added to the manuscript, as also suggested by R#1 (lines 53 and 447-454). As already stated, *Yamanasaurus* is far away from NW Argentina and SW Brazil, and is considered a saltasaurine titanosaurian. Thus, the new finding does not affect the known geographical distribution of Colossosauria.

RESULTS

Systematic paleontology

Systematic descriptions are appropriated and arguments to erect new taxa are convincing. Since I'm not an expert in titanosaurian taxonomy I have very little to say in this section:

7/ Line 75: If authors want to be consistent along the text and use American English, then they should use paleontology instead of palaeontology.

Reply: We have to keep it in British English.

8/ Line 144: The figure 2 caption indicates that "circled number correspond to

synapomorphies in the text”. However, the number 1 is marked in Fig2a, which corresponds to a cervical vertebra (C12), while the first synapomorphy, according to the text, is referred to middle dorsal vertebra (D6-D7) as correctly indicated in Fig2d. The miss-indication on figure should be emended.

Reply: We emended it.

9/ Line 223: “centra **is** spongy” change for “centra **are** spongy”

Reply: The subject is the tissue and not the centra (line 218).

10/ Lines 295-302: The division of the articular condyle of the quadrate is a very peculiar character indeed. I do not doubt of the validity of this character, but it is difficult to tell just from a picture; could you provide a better (perhaps more contrasted) picture? Can you ensure that this feature is not produced by taphonomic means? When comparing this character, what is the condition of the articular process in South American titanosaurs *Tapuiasaurus* and *Sarmientosaurus*?

Reply: We addressed this issue by introducing the following modifications on the original Fig. 3: a) We split the pictures in two new figures, one for the axial skeleton and one for the appendicular skeleton; b) we enlarged the pictures of the cranial bones in both lateral and ventral views; c) we added an interpretive drawing, since the preservation makes it difficult to observe the material. Regarding *Tapuiasaurus* and *Sarmientosaurus*, both are articulated. On the other hand, there is no specific information on this character in the publications and we have not made direct observations on these taxa.

Fig. 3.

Fig. 4.

11/ Lines 302-307: I am unable to recognize the quadratojugal in Fig.3a.

Is it like I tried to indicate in the figure?

Even if the quadratojugal extends more backward, how can the authors infer the projection of the anterior process of the bone? This region is not preserved in the figured specimen.

Reply: We addressed this issue in the last point.

12/ Lines 374-375: In addition to the current comparison, the deltopectoral crest is also markedly expanded distally in *Dreadnoughtus* and *Opisthocoelicaudia*, just if authors want to include this information to the description.

Reply: We added the information (now line 354).

13/ According to Fig. 1 both *Punatitan* and *Bravasaurus* preserve rib elements. One significant synapomorphy among titanosaurs is the occurrence of pneumatic foramen at the proximal end of the dorsal rib (see Mocho et al 2019). Is this feature present in the new taxa?

Mocho, P., Pérez-García, A., Martín Jiménez, M., & Ortega, F. (2018). New remains from the Spanish Cenomanian shed light on the Gondwanan origin of European Early Cretaceous titanosaurs. *Cretaceous Research*. doi:10.1016/j.cretres.2018.09.016

Reply: Ribs, as well as haemal arches are not particularly relevant for ingroup comparisons and we have a word limit for this paper. We provide several characters that support the titanosaurian affinities of *Punatitan* and *Bravasaurus*. Considering the limited space available we will publish that information elsewhere along with detailed osteology of both new taxa.

14/ Lines 389-394: Assuming that BM estimation is correct, 2.9 tones is a really low value for a titanosaure. The purported “dwarfs” titanosaurs *Ampelosaurus* from France and *Lirainosaurus* from Spain are estimated to be 2.5 and 1.8 tones respectively (see Sup. Excel file of Bensot et al. 2012). I realize that it is not the main scope of the paper, but I think that, apart of solely mentioning that *Bravasaurus* is the smallest colossosaurian yet recovered (lines 479-480), authors should propose some hypothetical scenario that helps to explain the occurrence of such small “titano” among colossal beast.

Reply: Yes, indeed, it is a low BM value. Although the information provided by the vertebrae is convincing regarding the adult size of *Bravasaurus*, we consider that it would be a bit premature to venture possible scenarios that explain its size. In addition, the text would extend into something that, as the reviewer clarifies, is far from the main scope of the paper. We added few sentences comparing *Bravasaurus* with dwarf sauropods from Europe (lines 372-375 and 466-468).

Phylogenetic analysis and biogeographic distribution

The phylogenetic results are convincing and help to depict a remarkable paleobiogeographic pattern. It is noteworthy the huge effort that the author made in recoding several taxa, which sure help improved the final results. However, I have some comments for this section:

15/ Although Bremer support index is provided in de Supplementary Material, I miss the report of other important phylogenetic indices, such as RI and CI. This information is useful to give an idea of the “robustness” of the results.

Reply: We added a new paragraph in the “Phylogenetic analysis” section of the Supplementary Information. It includes RI and CI values.

16/ Fig.4 I recommend to somehow highlighting the position of both *Punatitan* and *Bravasaurus* in the phylogeny, just to make visually easier its location.

Reply: Done.

17/ Since TNT is unable to establish time calibrations, to me, it remains unclear how authors established the time of divergence (nodes) between groups/taxa in the phylogeny. Did you consider any specific ratio of speciation/evolution? Did you place them according previous studies? This part should be improved given that it should affect parts of the discussion (Lines 481-493).

Reply: It seems that the term “time calibrated phylogeny” led to a wrong interpretation. We did not establish times of divergence between groups. The phylogenetic tree only contains occurrences of each taxon, according to published data. We modified the figure caption.

18/ I have to admit that I am not familiar with the paleogeographic characteristics of South America at the end of the Cretaceous (Campanian-Maastrichtian), and therefore I have some questions about this topic and how author established the criteria to conduct the paleobiogeographic analyses.

Reply: We have not carried out any specific palaeobiogeographic analysis. According to the palaeogeographic maps from Scotese and Wright (2018; see Fig. 1a of the manuscript), by the Late Campanian, there were large transgressions from the Pacific and Atlantic oceans that covered lowland territories of South America. The available information suggests that Brazil and Patagonia remained connected throughout the Cretaceous, so there would appear to be no limitations to the dispersal of titanosaurs. However, some studies hypothesize possible geographic barriers to explain the absence of saltasaurines in Brazil.

19/ If I am right, authors establish a “current” latitudinal gradient in order to define the location of different titanosaurs taxa. But, does the geographic position (latitude/longitude) of South America have remained invariable for the last 80-66 Ma? If the answer is yes (or almost negligible) then I agree that the current location of the fossil sites can be assumed as the same as that from 70 Ma. Otherwise, if the South

American continent experienced any significant displacement on the lithosphere during the last millions of years, then the geographic position should be corrected and the paleogeography analysis re-run considering past latitude/longitude locations.

Reply: The reviewer is right; we used a modern latitudinal gradient. Using palaeolatitudes does not represent a significant difference because South America moved more or less homogeneously since the Cretaceous. Much of that continental drift was due to the opening of the Atlantic Ocean, and therefore, the movement was mainly longitudinal. The latitudinal data barely vary between 3 and 5 degrees of latitude (data now included in Supplementary Information, Table S4). We modified the data and former Fig. 4 (now Fig. 5) so that they fit the palaeolatitude values. As you could see below, at this scale, the colour differences between the previous and new figure are minimal. Palaeolatitudes were calculated using the open source software GPlates (Seton *et al.*, 2012; Müller *et al.*, 2018).

As already stated, we didn't run a specific palaeobiogeographic analysis. The phylogenetic tree shows the occurrence of each taxon using a colour code referred to their palaeolatitudinal position (Supplementary Information, Table S4). The colour coding facilitates the visualization of latitudinal differences between clades of derived titanosaurs. Thus, at a glance, it is possible to identify that the Lognkosauria, for example, is a strictly Patagonian clade, while *Bravasaurus* has closer affinities with taxa that inhabited lower latitudes in Brazil.

We modified the “Latitudinal position of South American taxa” section of the Supplementary Information accordingly.

QSD Nesting Site

20/ Despite its remarkable significance, it seems that this section is somehow off place and disconnected from the main porpoise of the study, which – as far as I could understand – seems to be focused on provide evidence of the wide geographic distribution of the clade Colossosauria.

Reply: We regret that the section on the nesting site has seemed disconnected from the central purpose of the paper. This might be related to the difficulty in assigning the eggs to a clade within Titanosauria. We made some modifications to the text to better contextualize the results.

21/ However, since dinosaur eggs and reproduction is my main topic of study, I have some comments for the authors about this section.

22/ By contrasting the information provided in both the main text and SI, I found some problems to set a clear picture of the occurrence of titanosaure eggs in terms of geographic and stratigraphic distribution. Following I list the three main issues that I found:

1) Stratigraphic distribution

23/ The distribution of the 3 egg-bearing strata needs some clarification. In the main text it is mentioned that they outcrop within a 30 m thick sequence, which I suspect that it refers to the “grey interval” compressed between 45 and 70-75m above the base of Ciénaga del Río Hueco Fm. (according to Fig S1b), although no indication is provided in this regard. On the other hand, in the Supplementary Information it is said that they are distributed in a 5 m thick interval, that according to the text it is located 50-65 m above the base of the formation. First, that is actually 15 m interval not 5m, and second the interval is located around 60-65m from the base according to Fig. S1b.

So, what is the real stratigraphic distribution of the 3 egg-bearing levels? Please, improve this part.

Reply: The 30 m thick sequence referred in the main text is the stratigraphic interval of laminated siltstones and mudstones that contains the titanosaurian eggs. The three egg-levels, clearly separated stratigraphically, occur in a 5 m interval within this sequence. The reviewer is right; there was a typing error in the Supplementary Information, which we have corrected. To avoid confusion we modified the main text and Supplementary Information:

Main text (lines 404-406): The egg-clutches and eggshells are included in an interval of floodplain deposits in at least three distinct but closely spaced horizons at 59.2, 62.8 and 63.9 m above the base of the unit (Supplementary Fig. 1b).

Supplementary Information (292-302): Egg shells and egg clutches were found in at least three distinct but closely spaced horizons in a 5 m thick interval located 59-64 m above the base of Ciénaga del Río Huaco Formation (Supplementary Fig. 1b).

24/ In lines 429-430 it is stated that eggs and eggshells occur at different (stratigraphic?)

positions within the floodplain depositional sequence and, therefore, it is likely to be interpreted that they represent several nesting events.

However, when the authors say “several nesting events”, would they like to mean “three”, as the number of strata? How can the authors establish the number of nesting events?

Additionally, could it be that eggs with different positions within the profile could represent eggs of the same clutch but in a distinct internal location (in line 441 it is mentioned that eggs can occur in two-row arrangement) rather than different events?

Reply: The egg-bearing horizons at QSD are 59.2, 62.8 and 63.9 m above the base of Ciénaga del Río Huaco Formation. Within each level there are small vertical variations between different egg accumulations, and the eggshells often appear scattered at intervals several tens of centimetres thick. The floodplain sediments usually show evidences of pedogenesis. Soft sediment deformation and dislocation are frequent and we have documented them in QSD. As already documented for Auca Mahuevo (Jackson et al., 2013), the three egg bearing levels at QSD cannot be interpreted as three “nesting seasons”. Each one could constitute a time-averaged assemblage. We modified the text to clarify this.

2) Lateral extension of the nesting site

25/ The way in which the information is provided in the main text (lines 425 to 427) seems to suggest that the accumulations of eggs can be traced laterally over three kilometers, which would imply a humongous nesting area.

However, information from Supplementary Information seems to point that it is the stratigraphic interval (lithology/strata) the one that can be traced for more than 3 km, which leads a total different interpretation.

Thus, I suggest clarifying this point: Where egg remains recovered along 3 km? Or were they found in a single site (spot)?

Reply: The egg bearing strata are exposed along 3 km, and eggshells have been regularly found over more than 2 km of this stratum so far. We modified the main text (lines 410-413), see answer to the next comment (n°26).

26/ In addition of all of the above, it would be useful to provide the number of clutches, eggs, and eggshells discovered in the new nesting site, just to give a clear idea of the abundance and richness of the locality.

Reply: We completed the text as follows:

QSD Nesting Site. We documented three egg-bearing levels in the lower section of Ciénaga del Río Huaco Formation at QSD. The egg-clutches and eggshells are included in an interval of floodplain deposits in at least three distinct but closely spaced horizons at 59.2, 62.8 and 63.9 m above the base of the unit (Supplementary Fig. 1b). Fossil-bearing rocks are siltstones and sandy siltstones with horizontal lamination and graded and massive bedding that form thin tabular sheets, extending for tens to hundreds of meters. Some layers contain calcium carbonate as root encrustations and small soil nodules. This fossiliferous layer is laterally traced over more than three kilometres and the egg-clutches and

eggshells (CRILAR-Pv 620-621) are exposed regularly all along it. Nineteen egg clutches were spotted, one with up to 15 sub-spherical eggs, arranged in two superposed rows.

3) *Time occurrence*

27/ Authors seem to be somehow conditioned for the Auca Mahuevo nesting site, and omit many other evidences of “nesting site fidelity” in titanosaure sauropods. Additional examples of nesting recurrence in a single site can be found in both South America (see Salgado et al., 2007, 2009) and Europe (Sellés et al., 2013, 2017). I recommend including those additional references.

Salgado L, Coria RA, Ribeiro CM, Garrido A, Rogers R, Simón ME, Arcucci AB, Rogers KC, Carabajal AP, Apesteguía A, et al. 2007. Upper Cretaceous dinosaur nesting sites of Río Negro (Salitral Ojo de Agua and Salinas de Trapalcó-Salitral de Santa Rosa), northern Patagonia, Argentina. *Cretaceous Res.* 28:392–404.

Salgado S, Ribeiro CM, García RA, Fernández MS. 2009. Late Cretaceous Megaloolithid eggs from Salitral de Santa Rosa (Río Negro, Patagonia, Argentina): inferences on the titanosaurian reproductive biology. *Ameghiniana.* 46:605–620.

Sellés AG, Bravo M, Delclòs X, Colombo F, Martí X, Ortega-Blanco J, Parellada C, Galobart A. 2013. Dinosaur eggs in the Upper Cretaceous of the Coll de Nargó area, Lleida Province, south-central Pyrenees, Spain: oodiversity, biostratigraphy and their implications. *Cretaceous Res.* 40:10–20.

Sellés AG, Vila B, Galobar, 2017. Evidence of Reproductive Stress in Titanosaurian Sauropods Triggered by an Increase in Ecological Competition. *Scientific Reports* volume 7, Article number: 13827.

Reply: The term nest site fidelity is frequently used to describe nesting of modern reptile and bird species. In the literature, it is often used as a synonym for breeding-site philopatry and refers to the return of an individual or group of individuals of a species to the same location to breed.

As the reviewer clarifies, there are numerous sites in the world with multiple egg-bearing levels. However, in most cases, systematic allocation relies on parataxonomic comparisons. This nomenclatural system can obscure phylogenetic relationships (e.g., Grellet-Tinner et al., 2012; Jackson et al., 2013; Mikhailov 2019). In contrast, only a few sites such as Auca Mahuevo (Argentina), Totești (Romania) or Dholi Dungri (India) preserve embryonic remains *in ovo*, which confirm their assignment to Titanosauria. These sites represent, therefore, the most reliable source for morphological comparison within the framework of phylogenetic systematics. At Auca Mahuevo there are four egg-layers, three of which could correspond to the same species of titanosaurids (Hechenleitner et al 2015). We appreciate the suggestion to incorporate more citations. However, since we have already exceeded the maximum references for this journal, we try to keep those that could be more relevant for comparison.

28/ Line 432-433: It seems that “Auca Mahuevo” is repeated twice within the same

phrase.

“as reported for *Auca Mahuevo* the world-famous *Auca Mahuevo* nesting site, in the Argentine Patagonia”

It should be: “as reported for the world-famous *Auca Mahuevo* nesting site, in the Argentine Patagonia”

Reply: Sorry for the mistake. We modified this paragraph.

29/ Line 440: Authors starts the identification of QSD eggs mentioning that they share many features with those discovered in the Auca Mahuevo nesting site. But in my opinion the description of the QSD eggshell is so general that such features also agrees with several egg-types from India, Spain, France, Romania, Brazil, and several site from Argentina (i. ex. Salitral Rosa, Salitral Oje de Agua, Salitral Moreno, or Neuquen City). Thus, I suggest developing this section a little more. Apart from the shell thickness, shell unit shape, and pore canal system, authors should focus the new description and discussion on two additional features: 1) shape and contact between adjacent shell units, and 2) H:W ratio of the shell units.

1) shape and contact between adjacent shell units.

According to the most recent reviews on megaloolithid eggshells (see Fernández and Koshla, 2017 for further details) the way in which adjacent shell units growth allows to distinguish to main groups of megaloolithid eggs: *Megaloolithus*-with well-defined and straight edges; and *Fusioolithus*-with fused edged.

Eggs from Auca Mauevo have been included in the later group (Fernández and Koshla, 2017), while those from QSD site seems to posse well-defined shell margins (personal interpretation based on Fig.5d). I recommend the authors to take a look at the samples from the studies of Salgado et al (2007).

2) H:W ratio of the shell units.

This feature is commonly used to discriminate between “oospecies” of megaloolithid eggs (see Fernández and Koshla, 2017).

It seems that in QSD shells this values is about up to 2.5:1 or so, which is much closer to the specimens from Neuquen City (*Megaloolithus jabalparuensis*) than that of Auca Mahuevo (*Fusioolithus bahensis*).

Fernández Mariela S. & Khosla Ashu (2014): Parataxonomic review of the Upper Cretaceous dinosaur eggshells belonging to the oofamily Megaloolithidae from India and Argentina, Historical Biology: An International Journal of Paleobiology, DOI: 10.1080/08912963.2013.871718

Reply: As this is a short paper, the comparisons cannot be as extensive as in other journals. However, being brief does not mean that they are not exhaustive. The comparisons we presented in the previous version went straight to the point, avoiding comparisons with eggshell records of putative titanosaurs which do not look too much alike. We do not agree with the reviewer’s opinion: “...description of the QSD eggshell is so general that such features also agrees with several egg-types from India, Spain, France, Romania, Brazil, and several site from Argentina (i. ex. Salitral Rosa, Salitral Oje de Agua, Salitral Moreno, or Neuquen City).” The description “agrees” with several records around the world, and this makes sense as all them probably belong to

sauropod dinosaurs. However, among such diversity, several parameters such as the shell thickness, the diameter of the ornamentation, the size of the eggs and the morphology of the pore canals, show remarkable similarities with the eggs from layers 1-3 of Auca Mahuevo.

In the present version of the manuscript we extended the comparisons with confirmed records of titanosaurian eggs from India and Romania (e.g., Wilson *et al.*, 2010; Grellet-Tinner *et al.*, 2012) as well as some records that provide consistent information, such as those from Spain. The Brazilian record is fragmentary, but previous comparisons show strong similarities with the Auca Mahuevo site (Grellet-Tinner and Zaher, 2007). We added this information to the discussion.

We avoid the description and comparisons of the characters suggested by the reviewer because of the following reasons:

1) *Shape of the eggshell units*: It can vary between samples of the same site (personal observations EMH and LL). Some of these variations are related to diagenetic processes, both dissolution and recrystallization (e.g., Grellet-tinner *et al.*, 2010; Mikhailov, 2019). Regarding the apparent differences between the parataxonomic families mentioned by the reviewer, Fernández and Khosla (2014, p. 10) state:

“The shell units [of Fusioolithidae] are fan shaped similar to the eggs of oofamily Megaloolithidae but it differ in the nature of the eggshell units in which they are partially fused.”

In addition to vagueness in definition, the authors do not give nor suggest reasons that could explain such variations in terms of biological adaptation. As mentioned above, parataxonomy does not bring an appropriate framework for taxonomic identification, which is the main purpose here.

2) The H:W ratio of the shell units does not provide new relevant information. As we mentioned in the main text, titanosaurian eggshells are mono-layered, which means that the shell units make up the total thickness of the eggshell itself. They even shape the characteristic ornamentation, known as nodular. It would be risky to measure the width of the shell units in a radial section (as the reviewer suggests), since this section would hardly show a perfect radial cut of the units themselves. Furthermore, the thin section would have to provide enough perfect radial sections to be able to obtain a statistically significant value (Fig. 6d of the main text, illustrates the possible size variations for the shell units). Tangential sections are needed to measure the diameter of (many) shell units, and the solution for this is a bit tricky. As the diameter of the shell units increases outward, Fernández and Khosla (2014), for example, measure it on the external ornamentation. In other words, H equals shell thickness and W equals node diameter. Both data are available in the main text.

30/ Line 440 and 494: I believe that, by following my recommendations and processing a little bit the oological information, author should be able to extract some additional paleobiogeographic conclusions that may help to support their hypotheses.

Moving to the discussion section reserved for QSD eggs (starting from Line 494), this part must be largely improved.

Reply: Thank you. We followed the suggestions and modified the argument in the discussion of the revised manuscript.

Reviewer #3

GENERAL COMMENT

The paper is interesting and the materials are important. All comments are made via track changes in the attached document. I combined the supplemental info and main text into one document. More justification needs to be present in the main text; as it stands, so much is relegated to the supplemental information, the main paper doesn't stand on its own. The phylogenetic analysis unfortunately doesn't appear to have been very carefully done. I did not go through each and every character (let alone each character scoring), but looking over the list even briefly, I found a duplicate (copied and pasted) character, typos, and potential overlap among characters put forth as disparate. These issues will not give the reader confidence in the results of the phylogenetic analysis, if it is left in. I don't think a phylogenetic analysis is necessary, you could just note synapomorphies present in the material based on previous analyses. This would free up room for describing the provenance, age, and taphonomy of the specimens, all of which are severely underreported in the main text (but present in nice detail in the supplemental information). In sum, I think these are very important fossils and congratulate the authors for the huge undertaking involved in bringing them to light, but I feel that there are substantial issues that prevent publication of the manuscript in its present form.

Reply: We tried to address all the comments from the reviewer, but several of them pointed for a different kind of manuscript, which is not the case we have here presented. Most of the palaeontological and geological information in the present manuscript, including two titanosaurian sauropod species and a nesting site, are new. This means there is a lot to be said for contextualizing the reader in a short article. Therefore, we must transfer a lot of relevant data to the Supplementary Information. We conducted a careful phylogenetic analysis, which based using one of the most recent and complete phylogenetic data sets available (published in the Journal of Systematic Palaeontology). We provide detailed explanations regarding possible mistakes and duplicate characters below. We ran the phylogenetic analysis to understand the position of both new species in the context of the derived Titanosauria of South America, not as taxonomic support. Geological and taphonomic data are also essential for any new palaeontological record, but they are not central to the present investigation. We provide more geological data within the “nesting site” section of the Results.

ABSTRACT

Line 26_South American titanosaurs played a key role in the study of the evolution of...

R#3: Replace “played a key role in” with “were central to”

Reply: Modified.

Line 33...whereas the body mass of *Bravasaurus arrierosorum* gen. et sp. nov. is reminiscent of some dwarf titanosaurs in Europe.

R#3: Replace “the body mass of”, “reminiscent of”, and “in”.

Reply: We deleted the phrase as the abstract should be no longer than 150 words.

Line 35...this clade achieved a homogeneous distribution...

R#3: Replace “homogeneous” with “widespread”.

Reply: Same as the above.

Comment 1 the “Site” is not named in the abstract yet...what site? What constitutes a “site”—is it one quarry, one área, one región, etc? Be specific in how close these skeletons and eggs were found to one another, both spatially and stratigraphically.

Reply: The reviewer is right. We modified the abstract and now mentions the locality in line 31.

Comment 2 how far apart are these levels? how thick is each level?

Reply: The text of the abstract must be short. We cannot add more information, sorry.

INTRODUCTION

Line 45...attempted to establish paleobiogeographic links between these regions^{6,7}, although there are remarkable faunistic differences...

Comment 3: “Since” “because” or “finding” would seem to make more sense to me than “although”.

Reply: We disagree. “Since”, “because” or “finding” would change the meaning of the sentence.

Comment 4 correct me if I am wrong, but didn’t *Saltasaurus* live after the seaway had mostly retreated?

Reply: According to the palaeogeographic maps from Scotese and Wright (2018; see Fig. 1a of the manuscript), by the Late Campanian, there were large transgressions that covered lowland territories of South America. The seaways started retreating by the Maastrichtian.

Comment 5 it would be good to put here, upfront, the geographic and stratigraphic extent of the new locality. The way the main paper is written it sounds like the two new taxa were found at the same site; instead in the supplemental information it is stated that they are separated by ~150m stratigraphically.

Reply: The geographical (Quebrada de Santo Domingo locality in the Andes of La Rioja, NW Argentina) and stratigraphic (Upper Cretaceous red beds) locations are clarified in the “horizon and type locality” section of each taxon description (lines 91-93 and 250-252). We do not see that the main text is confusing regarding the origin (locality, quarry) of the two new taxa. The “horizon and type locality” section for each species accounts for their stratigraphic difference, just like what is stated in the Supplementary Information. We added a few words (lines 68-71) to avoid confusion: “We recovered three partial skeletons that belong to two new species of derived titanosaurian dinosaur species (Fig. 1c,d), and a new titanosaur nesting site, in different stratigraphic levels of the Ciénaga del Río Huaco Formation. Moreover, we found

titanosaurian egg-clutches and eggshells in an intermediate stratigraphic position, distributed in three bearing levels.”

RESULTS

Lines 100-101_4) extra-fossa ventrally to intersection of pcp1 and apcd1 in D6-D7*

Comment 7_ could this be interpreted as a divided fossa, and then named using the nomenclature for fossae proposed by Wilson et al. 2011?

Reply: We replaced the term “fossa” by “depression”. There is no specific landmark for naming this structure according to the nomenclature proposed by Wilson et al., (2011).

Comment 8_ “cervical” isn’t a noun, so please make sure you say “vertebra” after this and other positional adjectives throughout.

Reply: The reviewer is right. We corrected it throughout the main text.

Comment 9_ be more specific—based on what anatomical features? the parapophyses and diapophyses? shape of neural spine? centrum?

Reply: We added information in parentheses at the end of the sentence:

(e.g., relative position of parapophysis and diapophysis, orientation of neural spine).

Comment 10_ eyes come in many shapes. Do you mean “posteriorly tapering” or “anteriorly rounded and posteriorly acute” or some similar term?

Reply: Right, we modified the sentence:

... pleurocoels that have tapering, acute caudal margins.

Comment 12_ again, change to “dorsal vertebrae” throughout.

Reply: Done.

Comment 13_ more detail is warranted here...how many vertebrae are preserved, at a minimum? What are the overall proportions of the sacrum? Some further information must be available from excavation photos, right?

Reply: The referred sacrum is still within a plaster jacket. Little was visible in the field: a supraspinous rod and the tips of two neural spines. We think that excavation photos would not provide significant information for the present manuscript. We modified the few sentences for clarification:

The still unprepared sacrum of *Punatitan* is incomplete and will be described elsewhere. However, it was possible to observe an ossified supraspinous rod placed over the preserved neural spines (two or more).

Comment 14_ change “caudal” to “caudal vertebra” or “caudal vertebrae” throughout

Reply: Done.

Comment 15_ careful with use of “high” vs “tall”

Reply: We corrected it throughout the manuscript.

Lines ...change backwards along the vertebral column posteriorly from projecting laterally to projecting posterolaterally projected.

Reply: Modified as follows:

...gradually change from laterally to posterolaterally projected along the vertebral column.

Comment 16...define this here.

Reply: Done.

Comment 17_this seems like an odd phrase, since this is a description

Reply: Deleted.

Comment 19_please add in the following:

- tree length
- in the main text, please state how many additional steps are required to make *Bravasaurus* and *Punatitan* sister taxa, in either of their inferred positions.
- please list decay indices for all nodes in a table in supplemental information

Reply: We moved the tree length data to the Supplementary Information. We do not indicate how many additional steps are required to make *Bravasaurus* and *Punatitan* sister taxa in the main text because their strong taxonomic differences are marked along the text. Hence, that information is not relevant for the main purpose of the manuscript. If we apply the suggestion, there could be two possible scenarios: a high or a low number of steps. A high value would simply support the separation in two species. A low estimate of steps would help to justify that the two species could be closely related but in any case they could be considered the same. The requested decay indexes, Bremer support, RI and CI, are now available in the Supplementary Information.

Comment 20_much more detail is needed here in the main text. How thick is each egg-bearing level? How far apart is each stratigraphically? How do they relate to the bone-bearing layers for the new titanosaurs? They extend laterally more than three kilometers...three square kilometers? Is that just how far was surveyed, or is it known that that is where the extension ends? Lithologically, what are the rocks that host the egg-bearing layers? A stratigraphic column is needed.

Reply: Information regarding the stratigraphic position of the egg-bearing level is now provided in the main text (lines 404-413) and the Supplementary Information (lines 159-177). The relation with the new titanosaurs can be deduced from the results, as we already given the stratigraphic position of each taxon, as well as the egg-bearing levels. In addition, the data is also provided in the text of the Supplementary Information and Supplementary Fig. 1. Now, we briefly mention lithological information about the host rocks in the main text (lines 406-411). More information can be found in the Supplementary Information. The stratigraphic column is in Supplementary Fig. 1. It seems that the reviewer had some issues with the figures of the Supplementary Information (see below). Now we provide it in .pdf format.

Comment 21_stated as 5 meters thick in supplemental info

Reply: We modified the sentence in the main text.

Lines 447-450_For 40 years, *Saltasaurus* remained as the only well represented sauropod taxon for this region.

Comment 22_No; D'Emic and Wilson (2011) provided evidence that *Neuquensaurus* is also present at El Brete.

Reply: The record of a putative *Neuquensaurus* specimen consists of a poorly preserved sacrum, in a place in which there are several specimens of *Saltasaurus*. Although it is not relevant for this paper, we slightly modified few sentences and added the requested reference.

For 40 years, *Saltasaurus* remained the only well represented sauropod taxon for this region. *Saltasaurus* is closely related with the Patagonian *Rocasaurus* and *Neuquensaurus* (also has a putative record in Salta(D'Emic and Wilson, 2011)), as well as the recently discovered *Yamanasaurus*(Apesteguía *et al.*, 2020), from Ecuador.

Lines 480-482_Besides, some taxa recovered within Rinconsauria are often included within Aeolosaurini, a group of titanosaurs with doubtful affinities(Martinelli *et al.*, 2011).

Comment 23_Affinities to what?

Reply: We slightly modified the sentence that now reads:

Besides, some taxa recovered within Rinconsauria are often included within Aeolosaurini, a group of titanosaurs with unstable interspecific phylogenetic relationships (Martinelli *et al.*, 2011)

Lines 502-503_Further material of *Bravasaurus* would help to evaluate whether the space at the pelvic girdle was large enough to allow the passage of these eggs.

R#3: whether ~~the space~~ at the pelvic girdle was large enough to ~~allow~~ accommodate the passage of these eggs.

Reply: We simplified this sentence as it would be quite confusing. It now reads:

Further materials are required to evaluate each scenario.

METHODS

Comment 24_the new titanosaurs' long bones should really be histologically sampled to assess somatic maturity.

Reply: The inference of somatic maturity in the holotype of *Bravasaurus* is supported by data provided in several publications (already cited in the main text). Future histological sampling in the context of a more specific study could help confirming this inference.

Line 555_We added five characters and modified several scorings.

Comment 25_here please state that you added two characters from a previous study and created three new ones.

Reply: Sorry, we corrected the phrase and now it reflects what we had already written in the Supplementary Information.

We added five characters (three from previous studies and two new) and modified few scorings.

Comment 26_this seems like a mix of results and methods.

Reply: Modified (lines 559-562). We moved the number of steps, along with other information regarding of MPTs at different steps of the analysis, to the Supplementary Information.

FIGURES

Fig. 1_ it is confusing to use green and yellow for these silhouettes because green and yellow are used for the percentage regions in part A of the figure. So at first glance it looks like *Bravasaurus* is from Brazil and *Punatitan* is from La Rioja. I recommend using different colors.

Reply: We have changed the colours of the preserved bones of the titanosaurian species. Now both are red.

Fig. 2_part F only shows these vertebrae slightly larger than they are shown in part E. I would recommend cropping out the first and last vertebra shown in F and enlarging the rest.

Reply: We modified the figure.

Fig. 4_neither the caption nor the text mention why *Bravasaurus* has a dashed line and question mark joining it to the rest of the tree.

Reply: The first version of the manuscript states: Dashed line for *Bravasaurus* refers to its possibly older age (see Supplementary Information). In the new version we modified the figure. Now *Bravasaurus* shows the same temporal range than *Punatitan*, as variations in age cannot be confirmed. See comment 19 from R#2.

SUPPLEMENTARY INFORMATION

Comment 27_this image is completely black

Reply: Sorry, there might be a conflict with the figures included in the word text. We submit the Supplementary Information in .pdf format.

Comment 28_again, this image is completely black. with the next round of review please also upload the supplemental info as a pdf.

Reply: Same as comment 27.

Comment 29_I'd expect to see Phylogenetic methods more precisely laid out before results. What are the five new characters? What scoring changes were made, precisely? Why only parsimony?

Reply: We moved part from the method section of the main text to this section (207-213). We added a first paragraph explaining the workflow that we applied. The present paper is not focused on systematics. We introduced minimum changes into the matrix of Carballido et al. (2020), and they do not severely modify their. The authors provide an extensive discussion regarding the methods and limitations of their dataset.

Comment 32_the source for these data needs to be cited. is it from original literature? paleobiology database? elsewhere? also, is it paleolatitude or current latitude?

Reply: We incorporated more data for clarification. See comment 19 from R#2.

Comment 33_there are a very large number of changes, but no discussion or documentation of those changes. at a minimum, for each change you should cite the change's source, i.e., a publication and its figure number or pers. obs. with the author's initials who made the pers. obs.

Reply: For this paper, we made 169 modifications to the dataset provided by Carballido et al. (2020). This value represents 0.4% of the total data (422x98). Of these modifications, 80% were related to missing data that we change to character-states. The new information comes from publications (e.g. *Overosaurus*, *Rapetosaurus*) and personal observations (e.g. *Trigonosaurus*, *Baurutitan*, '*Aeolosaurus*'). We made 32 changes to the original scorings of Carballido et al. (2020); 16 of which represent modifications of the characters' definitions or their states. For instance, we split one state of character 300 into two, which causes variations in the scorings that do not reflect a different interpretation of the observed morphology. Among the other 16, three are changes to ambiguous states (e.g. 0 to 0&1), one to "not-applicable" and another to missing data. We explain the remaining 11 modifications in Supplementary Table 7.

Comment 34_It would be helpful to say “state 0 added” or “state 1 added” etc. as appropriate

Reply: Done.

Comment 35_“new character” implies you created it, but two are from a previous study, so please modify this text.

Reply: Corrected.

Comment 36_replace x with the arrow throughout

Reply: Done.

Comment 37_some of the genera are not italicized; please fix

Reply: Done.

Comment 38_duplicated character

Reply: Nope. The characters 58 and 59 are different:

58. Quadrate, articular surface shape: quadrangular in ventral view, oriented transversely (0); roughly triangular in shape or thin, crescent-shaped surface with anteriorly directed medial process (1).

59. Quadrate, articular surface shape: quadrangular in ventral view, oriented transversely or roughly triangular in shape (0); thin, crescent-shaped surface with anteriorly directed medial process (1).

Two characters with two states (different from each other) are the same than one character with three states, right?

Comment 39_don't follow what this means; wrong word choice

Reply: Sorry, it was just a misspelling. Corrected.

Comment 40_table 5 says this is modified to add a state but there are only two states listed. please fix.

Reply: The reviewer is right. We fixed the error.

Comment 42_how is this different from character 257? different portions of the column? what even is an “anterior-posterior caudal vertebra”?

Reply: Yes, different portions of the column.

Comment 43_racquets come in lots of shapes. please be more precise.

Reply: This character, created by Wilson (2002: character 152) and modified by Carballido et al. (2017: character 277), corresponds to the scapula. The state 2, “racquet-shaped” corresponds to taxa, such as *Nigersaurus*, *Rebbachisaurus*, which are not directly related with derived titanosaurians. Maybe the word is not the best one. However, we do not see the need to change it in this paper.

Comment 44_“notorious” means famous for a bad thing.

Reply: None of the authors are native English speakers, however, according to the Cambridge dictionary, prominent can also be considered “noticeable”:

Sticking out from a surface:

E.g. *She has a prominent chin/nose.*

Comment 45_what does “extremely” mean?

Reply: Several character lists use these terms in an effort to discretize a morphology that sometimes has continuous variations. Many times it is possible to understand how previous authors, like Wilson (2002) and Whitlock (2011), mean when using "reduced" and "extremely reduced", by observing the already scored material.

Comment 46_do you mean “anterior margin of neural spine located posterior to plan of anterior margin of postzygapophyses”?

Reply: Yes, we follow the definition provided by Salgado et al. (1997). Now we used “anterodorsal end of the neural spine” instead of “anterodorsal border...” since it seems to be confusing.

Comment 47_characters 257, 260, and 419 seem to be the same character.

Reply: We already explained the difference between characters 257 and 260 in a previous comment. Some derived titanosaurs like *Bonitasaura* and *Gondwanatitan* have middle caudal vertebrae with neural spines slightly inclined posteriorly (state 2 in character 257), but the anterodorsal end of the spines do not surpasses the anterior border of the postzygapophyses (state 0 in character 419). In contrast, *Saltasaurus* and *Rocasaurus* also show neural spines slightly inclined posteriorly (state 2 in character 257) and the anterodorsal end of the spines surpasses the anterior border of the postzygapophyses (state 1 in character 419). State 1 in character 419 is common to the South American saltasaurids.

Comment 48_do you mean expanded dorsoventrally? transversely? both?

Reply: We corrected the definition according to the suggestion.

REFERENCES

- Apestequía, S., Soto Luzuriaga, J.E., Gallina, P.A., Granda, J.T. and Guamán Jaramillo, G.A. 2020. The first dinosaur remains from the Cretaceous of Ecuador. *Cretaceous Research* 108: 104345.
- Borsuk-Białynicka, M. 1977. A new camarasaurid sauropod *Opisthocoelicaudia skarzynskii* gen. n., sp. n. from the Upper Cretaceous of Mongolia. *Paleontologia Polonica* 37: 5–64.
- Carballido, J.L., Scheil, M., Knötschke, N. and Sander, P.M. 2020. The appendicular skeleton of the dwarf macronarian sauropod *Europasaurus holgeri* from the Late Jurassic of Germany and a re-evaluation of its systematic affinities. *Journal of Systematic Palaeontology* 18: 739–781.
- D’Emic, M.D. and Wilson, J. a. 2011. New Remains Attributable to the Holotype of the Sauropod Dinosaur *Neuquensaurus australis*, with Implications for Saltasaurine Systematics. *Acta Palaeontologica Polonica* 56: 61–73.
- Fernández, M.S. and Khosla, A. 2014. Parataxonomic review of the Upper Cretaceous dinosaur eggshells belonging to the oofamily Megaloolithidae from India and Argentina. *Historical Biology* 27: 158–180.
- Grellet-tinner, G., Corsetti, F. and Buscalioni, A.D. 2010. The importance of microscopic examinations of eggshells: Discrimination of bioalteration and diagenetic overprints from biological features. *Journal of Iberian Geology* 36: 181–192.
- Grellet-Tinner, G. and Zaher, H. 2007. Taxonomic identification of the Megaloolithid egg and eggshells from the Cretaceous Bauru Basin (Minas Gerais, Brazil): Comparison with the Auca Mahuevo (Argentina) Titanosaurid eggs. *Papéis Avulsos de Zoologia* 47: 105–112.
- Grellet-Tinner, G., Codrea, V., Folie, A., Higa, A. and Smith, T. 2012. First evidence of reproductive adaptation to ‘island effect’ of a dwarf Cretaceous Romanian titanosaur, with embryonic integument in ovo. *PLoS ONE* 7: e32051.
- Martinelli, A., Riff, D. and Lopes, R. 2011. Discussion about the occurrence of the genus *Aeolosaurus* Powell 1987 (Dinosauria, Titanosauria) in the Upper Cretaceous of Brazil. *Gaea* 7: 34–40.
- Mikhailov, K.E. 2019. Conservative nature of biomineral structures as a challenge for the cladistic method of phylogeny reconstruction (illustrated by two groups of dinosaur eggs). *Paleontological Journal* 53: 551–565.
- Müller, R.D., Cannon, J., Qin, X., Watson, R.J., Gurnis, M., Williams, S., Pfaffelmoser, T., Seton, M., Russell, S.H.J. and Zahirovic, S. 2018. GPlates: Building a Virtual Earth Through Deep Time. *Geochemistry, Geophysics, Geosystems* 19: 2243–2261.
- Otero, A., Gallina, P.A., Canale, J.I. and Haluza, A. 2011. Sauropod haemal arches: morphotypes, new classification and phylogenetic aspects. *Historical Biology* 1–14.
- Poropat, S.F., Mannion, P.D., Upchurch, P., Hocknull, S.A., Kear, B.P., Kundrát, M., Tischler, T.R., Sloan, T., Sinapius, G.H.K., Elliott, J.A. and Elliott, D.A. 2016. New Australian sauropods shed light on Cretaceous dinosaur palaeobiogeography. *Scientific Reports* 6: 34467.
- Poropat, S.F., Upchurch, P., Mannion, P.D., Hocknull, S.A., Kear, B.P., Sloan, T., Sinapius, G.H.K. and Elliott, D.A. 2015. Revision of the sauropod dinosaur *Diamantinasaurus matildae* Hocknull et al. 2009 from the mid-Cretaceous of Australia: Implications for Gondwanan titanosauriform dispersal. *Gondwana Research* 27: 995–1033.

- Salgado, L., Coria, R.A. and Calvo, J.O. 1997. Evolution of titanosaurid sauropods. I: Phylogenetic analysis based on the postcranial evidence. *Ameghiniana* 34: 3–32.
- Seton, M., Müller, R.D., Zahirovic, S., Gaina, C., Torsvik, T., Shephard, G., Talsma, A., Gurnis, M., Turner, M., Maus, S. and Chandler, M. 2012. Global continental and ocean basin reconstructions since 200Ma. *Earth-Science Reviews* 113: 212–270.
- Wilson, J.A. and Upchurch, P. 2003. A revision of *Titanosaurus* Lydekker (Dinosauria - Sauropoda), the first dinosaur genus with a ‘Gondwanan’ distribution. *Journal of Systematic Palaeontology* 1: 125–160.
- Wilson, J.A., Mohabey, D.M., Peters, S.E. and Head, J.J. 2010. Predation upon hatchling dinosaurs by a new snake from the Late Cretaceous of India. *PLoS Biology* 8: e1000322.

Reviewers' comments:

Reviewer #1 (Remarks to the Author):

This is now publishable almost as is - I only have a few minor corrections:

Line 237: as those -> as are those.

Line 290: withribs -> with ribs.

Line 338: citation(s) needed for neural arch position on titanosaurs and Wintonotitan.

Line 371: no need to italicise proxy.

Lines 374 & 375: don't use 'has' for mass; maybe phrase as "The mass of the European Lirainosaurus was less than two tons, whereas that of the Argentinean Saltasaurus and Neuquensaurus were five and six tons, respectively."

Looking forward to seeing this published! Well done.

Reviewer #2 (Remarks to the Author):

I'm pleased to observe that authors took into consideration most of my comments. I'm particularly satisfied in noting that many sections of the text have been improved. The introduction is much clearer than the previous version, remarking the significance of Colososauria and QSD egg-site.

The new descriptions and comparison of fossil remains help to highlight the singularity of the new two titanosaurian species.

I also appreciate the modification and improvement of figures, specially Fig3.a. Now, the morphology and anatomical relationship of quadrate and quadratojugal can be perfectly observed.

I really thank the additional comments on the stratigraphic position of the egg-bearing strata that constitute the QSD nesting-site. With the new information I agree with the authors that the 3 egg levels cannot be interpreted as 3 "nesting season events". As a consequence, I approve the new description of the QSD titanosaurian nesting site and eggs as it is in the new version of the manuscript.

One last thing, I attach a reviewed version of the text marking "insignificant" typing errors.

In summary, I think that the manuscript can be published as it is in its final version.

Reviewer #3 (Remarks to the Author):

I cannot support publication of this paper in present form. If the space limitations of this journal limit the inclusion of key information in the main text and abstract, then this paper should not be published in this journal. If the authors cannot recognize that characters are duplicated, addressing the same anatomical variation with different wording, then they should not be publishing a phylogenetic analysis. Finally, the paper is fundamentally framed as an association between two "dispersed" species and a nesting site, when in reality these discoveries are spatiotemporally distinct (despite the abstract now misleadingly stating that these were all found "at the same locality." Unfortunately I recommend rejection of this paper.

Rebuttal letter

Here we provide further details regarding R#3's concerns.

Character duplication in the phylogenetic analysis

We made the pertinent clarification in both cases of alleged similar or duplicate characters in the first round of revision. Nonetheless, we here provide more data for the following characters:

a) Characters 58 and 59: Whitlock (2011) first created a single character (character 32) with three states. Mannion et al. (2012) split this character for the first time, into two characters with two states each (which is exactly the same that ordering the multistate character of Whitlock). Characters 29 and 30 from Mannion et al (2012) state:

Ch 29. [W32, modified] Quadrate, articular surface shape: quadrangular in ventral view, oriented transversely (0); roughly triangular in shape or thin, crescent-shaped surface with anteriorly directed medial process (1) (Whitlock 2011). Whitlock (2011) used a three-state multistate ordered character. Here, the same data are represented using two additive binary characters (character nos. 29 and 30).

Ch 30. [W32, modified] Quadrate, articular surface shape: quadrangular in ventral view, oriented transversely or roughly triangular in shape (0); thin, crescent-shaped surface with anteriorly directed medial process (1) (see character no. 29 above).

Carballido et al (2020) followed this same criterion (two characters with two states each), whereas for example, Tschopp et al (2015) used the single character (ordered), following Whitlock (2011). These options involve the same number of steps in the analysis and are therefore equivalent.

b) Characters 257, 260 and 419: The first two characters may look similar, although they apply for different (and immediately consecutive) regions of the tail. The orientation of the neural spines can vary along the caudal series. For example, in *Gondwanatitan* the posteriormost anterior and middle caudal vertebrae have the neural spine slightly directed posteriorly (ch. 257: state 2), and in the posterior caudals the neural spines are strongly directed posteriorly (ch. 260: state 2). In contrast, in *Patagotitan* and *Bonitasaura* the neural spines of the posteriormost anterior and middle caudal vertebrae are also slightly directed posteriorly (ch. 257: state 2), but in the posterior caudals the neural spines are vertical (ch. 260: state 0). Several other differences can also be observed in the dataset.

In addition, we added another character in our data matrix, taken from Salgado et al (1997), which allows discriminating the morphology of the neural spine of saltasaurids:

Ch 419. Middle caudal vertebrae, anterodorsal end of the neural spine located posteriorly with respect to the anterior border of the postzygapophyses: absent (0); present (1).

We suggest that this variation is related to the development of the neural spine. It is important to note here that the scorings are not the same for the three cited characters, among terminal taxa. Different scorings mean that the characters are not the same and that one is not (apparently) dependent of the other. An email of Dr. Jose Carballido with further discussion is attached.

Spatiotemporal association between the species

We found the holotypes of *Bravasaurus* and *Punatitan* separated (spatially) less than 300 m from each other in the outcrops of Ciénaga del Río Huaco Formation, at Quebrada de Santo Domingo. Both the main text and supplementary information clearly state that the stratigraphic gap between them is less than 150 m. Perhaps the expression “fossil assemblage” (line 62 of the main text) led to some confusion. In the manuscript, we do not propose that both species were contemporary. Previous geological studies assigned the Ciénaga del Río Huaco Formation to the Late Cretaceous. We also briefly discuss (in the supplementary information) about the possibility that the *Punatitan* strata belong to the overlaying Puesto La Flecha Formation (see pages 2 and 3 of the supplementary information). Such change, however, would not be relevant for the goal of this contribution. Currently, the available data allows us to limit the stratigraphic interval with dinosaur remains to the Late Cretaceous. Therefore, although *Punatitan* and *Bravasaurus* may not have lived at the same time, both are limited to an identical temporal lapse. Whether or not they were strictly contemporaneous does not affect in any way the conclusions of the present manuscript. In order to avoid any misunderstanding, in the revised version of the manuscript we replaced “new fossil assemblage” with “...new dinosaurs...” (line 62).

References

Mannion, P. D., Upchurch, P., Mateus, O., Barnes, R. N. & Jones, M. E. H. New information on the anatomy and systematic position of *Dinheirosaurus lourinhanensis* (Sauropoda: Diplodocoidea) from the Late Jurassic of Portugal, with a review of European diplodocoids. *Journal of Systematic Palaeontology* 10, 521–551 (2012).

Salgado, L., Coria, R. A. & Calvo, J. O. Evolution of titanosaurid sauropods. I: Phylogenetic analysis based on the postcranial evidence. *Ameghiniana* 34, 3–32 (1997).

Tschopp, E. A specimen-level phylogenetic analysis and taxonomic revision of Diplodocidae (Dinosauria, Sauropoda). (2015). doi:10.7717/peerj.857

Whitlock, J. A. A phylogenetic analysis of Diplodocoidea (Saurischia: Sauropoda). *Zoological Journal of the Linnean Society* 161, 872–915 (2011).

Esteban Martín Hechenleitner <emhechenleitner@gmail.com>

Sauropod phylogeny dataset 2020 - some questions

José Luis Carballido <jcarballido@mef.org.ar>
Para: Martín Hechenleitner <emhechenleitner@gmail.com>
Cc: martin@conocet.gov.ar

1 de julio de 2020, 20:17

Hi Martin (and team). I presume you are working with some not Spanish speaker, so I will answer you in English (sorry if there is something not really clear).

Character 58. This character was initially proposed by Whitlock (2011:ch. 32). First he used it as one character with three states. After that the character was split into two characters with two states each (which is exactly the same that ordering the multistate character of Whitlock; see Mannion et al., 2011 the Dinheriosaurus paper). Not sure what your doubt about it, you can use it as you prefer (either ordered as a single character with three states or using two characters with 2 steps each; as most of the recently published phylogenies are doing). But note that will not be the same if you decide to unite these two characters in one character and not using it as additive. Tschopp et al (2015, in their phylogenetic paper) used as originally described by Whitlock and ordered. So, for the analysis is the same, just change the way in which you describe the character. All other analyses based on Whitlock, Mannion, and Tschopp used in one of the two ways, and some others incorporate it in their data sets (as I did).

Character 257 and 260 from my matrix and the new one that you want to include. Yes, you are right they sound as the same character but actually they are not the same. The orientation of the neural spine suffers a series of changes throughout the caudal series of many sauropods. In this case, the orientation of the neural spine. There are some sauropods which middle caudals shows a slightly posteriorly directed neural spine, but the neural spine is more strongly posterodorsally directed in posterior caudals (that is, the neural spine becomes more posteriorly directed throughout the caudal series), whereas in others the neural spine becomes more vertically oriented in posterior caudal vertebra than in middle caudals. I think that your new character is, in part, product of the orientation but also seems to be product of the length and development of the neural spine. Looking the matrix that you sent me and going back to my pictures and works on some titanosaurs looks evident for me that the characters are pretty similar but having some differences in their scores. So, it seems that with them you can help to better represent the morphological variation observed in the neural spine of these taxa. So, I am in agree with you in considering these two characters plus the one that you want to include in your work. In general, if you have some taxa with elements of the same position and you can score them differently for these characters means that are not the same and that one is not (apparently) dependent of the other. The character 257 and 260 was introduced by me together with Leo Salgado, and I remember that we discuss that extra variation was evident but was out of our scope in that moment.

Not sure if this helps you, but please let me know if you want to discuss this further. We can keep discussing it by phone, but in my opinion you can feel sure to use them.

Cheers

jose

El 30 jun. 2020, a las 17:21, Martín Hechenleitner <emhechenleitner@gmail.com> escribió:

[El texto citado está oculto]

REVIEWERS' COMMENTS:

Reviewer #1 (Remarks to the Author):

This manuscript is, and has for some time, been perfectly fine for publication. I recommend acceptance almost as is: the only change I recommend is to line 104: "...allowing distinguishing the new taxon from other titanosaurs." should be "...allowing us to distinguish the new taxon from other titanosaurs."

Reviewer #2 (Remarks to the Author):

After carefully reading the author's answered to my requests, I must say that I'm satisfied with the response and that I agree with all their arguments. Therefore, I have no additional comments or observations to make upon the study.

I specially appreciate the effort in modification the introduction, the addition of new figures, and the arrangement of the nesting-site section.

Overall, I think this is a very interesting study that should be considered for publishing in its current state.